# Wdr1 and cofilin are necessary mediators of immune-cell-specific apoptosis triggered by Tecfidera

Jesse R. Poganik [1,2], Kuan-Ting Huang[1], Saba Parvez[3], Yi Zhao[4], Sruthi Raja[1], Marcus J. C. Long [5✉] & Yimon Aye [1✉]

Despite the emerging importance of reactive electrophilic drugs, deconvolution of their principal targets remains difficult. The lack of genetic tractability/interventions and reliance on secondary validation using other non-specific compounds frequently complicate the earmarking of individual binders as functionally- or phenotypically-sufficient pathway regulators. Using a redox-targeting approach to interrogate how on-target binding of pleiotropic electrophiles translates to a phenotypic output in vivo, we here systematically track the molecular components attributable to innate immune cell toxicity of the electrophilic-drug dimethyl fumarate (Tecfidera®). In a process largely independent of canonical Keap1/Nrf2-signaling, Keap1-specific modification triggers mitochondrial-targeted neutrophil/macrophage apoptosis. On-target Keap1–ligand-engagement is accompanied by dissociation of Wdr1 from Keap1 and subsequent coordination with cofilin, intercepting Bax. This phagocytic-specific cell-killing program is recapitulated by whole-animal administration of dimethyl fumarate, where individual depletions of the players identified above robustly suppress apoptosis.

[1] Swiss Federal Institute of Technology Lausanne (EPFL), Lausanne, Switzerland. [2] Division of Genetics, Department of Medicine, Brigham and Women's Hospital, Harvard Medical School, Boston, MA, USA. [3] Department of Pharmacology and Toxicology, College of Pharmacy, University of Utah, Salt Lake City, UT, USA. [4] BayRay Innovation Center, Shenzhen Bay Laboratory (SZBL), Guangdong, China. [5] University of Lausanne (UNIL), Lausanne, Switzerland. ✉email: marcusjohncurtis.long@unil.ch; yimon.aye@epfl.ch

The mysteries of electrophilic drugs have lingered long in the minds of medicinal chemists, biologists, and ultimately physicians. Appending an electrophilic motif to a reversible inhibitor can confer significantly-improved pharmacological properties[1,2]. Conversely, electrophilic motifs are also associated with crippling off-target effects causing drug toxicity, withdrawal from the market, and in extreme cases, death[3]. It remains a formidable challenge to pinpoint the key functional targets (good and bad) of many electrophilic drugs, and even more so for particularly promiscuous electrophilic compounds. However, this step is necessary for optimizing therapeutic profiles around specific targets.

In lieu of accurate mechanistic analyses, many broad-specificity electrophilic candidates, approved or in clinical trials, are proposed to be polypharmacologic (i.e., boast various important targets)[4,5]. How much polypharmacology is attributable to, or even necessary for, efficacy remains unknown. This uncertainty applies to electrophilic natural products and drug candidates. We are thus left with a trove of reactive covalent drugs, drug candidates, and natural products, with little information on how these compounds function. Such confusion and inability to systematically address these questions limit our ability to improve drug efficacy and predict contraindications, and ultimately hinder approval and improving lives. A general method to help address the importance of the role of specific protein targets in electrophilic drug mechanisms would thus be very useful.

The approval in 2013 of the pleotropic electrophilic drug Tecfidera (DMF), a multibillion-dollar drug that treats relapsing multiple sclerosis (MS)[6] was a watershed moment in electrophilic therapeutics. However, this molecule is also a poignant example of our inability to understand the biology of electrophilic molecules. Critically, DMF is an ene-diester and hence is significantly more electrophilic than most approved electrophilic anti-cancer drugs, which contain enamides. Thus, unraveling the mode-of-action of DMF has proven to be particularly challenging, and there remain conflicting proposals. DMF functions to a significant extent through suppression of the immune response and inflammation that are the root causes of MS[7]. Immune-cell suppression is often ascribed to immune-cell-specific apoptosis, although mechanisms remain unknown[8,9]. Suppression of immune-cell maturation has also been proposed; one possible target in this vein is GAPDH[10]. However, there is no data pertaining to how precise target engagement of GAPDH by DMF affects immune cells. Proteomics approaches to identify targets following bulk administration of DMF (and analogs) to specific cells in culture have unearthed other interesting putative targets[11,12]. However, the ultimate conclusion of these proteomics efforts was that the targets so identified are insufficient regulators of drug-induced cell response. Unfortunately, these recent results have overall eroded confidence that the originally-proposed principal target of DMF, and a known cellular target of fumarate (from which DMF is derived), Keap1, is a relevant target.

In our opinion, the studies above and others in the literature trying to unravel DMF's mechanism(s) remain limited specifically because they are unable to precisely probe the functional consequences of "on-target" ligand-engagement in any way. Our recently developed on-target signaling interrogation tool, T-REX[13], that is compatible with zebrafish[14] ("Z-REX" hereafter), presents an opportunity to begin to address these questions. This method releases a controlled amount of a specific enone/enal-based electrophile to a preordained protein of interest (POI) with spatiotemporal control (Supplementary Fig. 1a). Thus, Z-REX has the ability to probe the functional consequences of on-target POI–electrophilic drug interactions, either directly or using a surrogate to mimic the drug-modified state of the POI, generating data that can later be evaluated using the non-modified drug. We

have developed ideal controls, where the specific electrophile is not shepherded to the target POI (i.e., non-targetable P2A Z-REX construct, Supplementary Fig. 1a bottom row), to account for variables that may be incurred upon T(Z)-REX that could lead to confounding outputs. Particularly for reactive electrophiles, such controls are essential, but are another aspect unavailable to other approaches.

We here leverage Z-REX to link measurable phenotypic outcomes in live fish following covalent ligand association with Keap1, which in our opinion remains the most likely protein sensor of DMF. We discovered a neutrophil- and macrophage-specific apoptosis pathway that is triggered by Keap1 engagement with some, although not all electrophiles examined. In the process, we discovered necessary protein players triggered into action as a result of Keap1-modification by the active electrophiles. Critically, deficiency of players in this pathway fully suppresses the phenotypes that accompany whole-animal DMF administration, which we demonstrate using targeted knockdown/knockouts in various models, including larval fish and mouse primary bone marrow-derived macrophages. The level of sufficiency and necessity observed is in stark contrast to limited functional contributions made by previously-identified DMF-responsive targets/pathways[10–12,15]. The coupling of mechanistic analysis leveraging the Z-REX electrophile toolbox with bulk administration of the electrophilic drug under investigation thus represents a versatile method to interrogate electrophilic drug-induced phenotypes. In the case of neutrophil/macrophage death, the phenotypes of whole-animal DMF administration are almost entirely ascribable to the specific pathway we identify.

## Results

**Chemotype-specific Keap1-modification suppresses immune genes.** We chose to study Keap1, a commonly-proposed target of electrophilic molecules, for 3 key reasons. (1) Keap1 is a major electrophile sensor, functioning in the antioxidant-response (AR) pathway of conserved importance[16]. (2) Necessity of Keap1 in mode-of-action of emerging covalent drugs remains untestable using conventional approaches because Keap1-knockout is not well tolerated[17] and misregulates critical electrophile-responsive pathways, significantly modulating pharmacokinetics[18,19]. (3) Keap1 regulates myriad cell responses beyond AR, but these cascades are uninvestigated vis-à-vis DMF mechanism. We further chose to perform experiments using Z-REX because Z-REX accounts for (2) and (3) by allowing examination of functional consequences exclusively attributable to Keap1-specific covalent modification, in the backdrop of largely-unperturbed embryos. Shown against a suite of controls (Supplementary Fig. 1a bottom row), Z-REX also poses no adverse effects on development or viability, for days after the experiment[14].

We first launched a Z-REX-coupled RNA-seq screen to identify genes whose expression is significantly changed following Keap1-specific electophile engagement[18] (Supplementary Fig. 1a top row). We took advantage of the available T(Z)-REX photocaged electrophile toolbox[13], whereby several of the native electrophiles delivered engage with Keap1 in cells. Our hypothesis was that at least one of these would mimic DMF-derived succination of Keap1. We chose to deliver 4-hydroxynonenal (HNE) and 4-hxdroxydodecenal (HDE) to Keap1 in larval fish and measured Keap1-modification-dependent transcriptomic changes in each case. The data from Z-REX—RNA-seq delivering HNE revealed that, relative to untreated embryos, embryos that had undergone Keap1-specific-HNE-modification (hydroxynonenylation hereafter) upregulated transcripts associated with Keap1/Nrf2-antioxidant-response (AR) signaling[19] [Fig. 1a, in green], validating

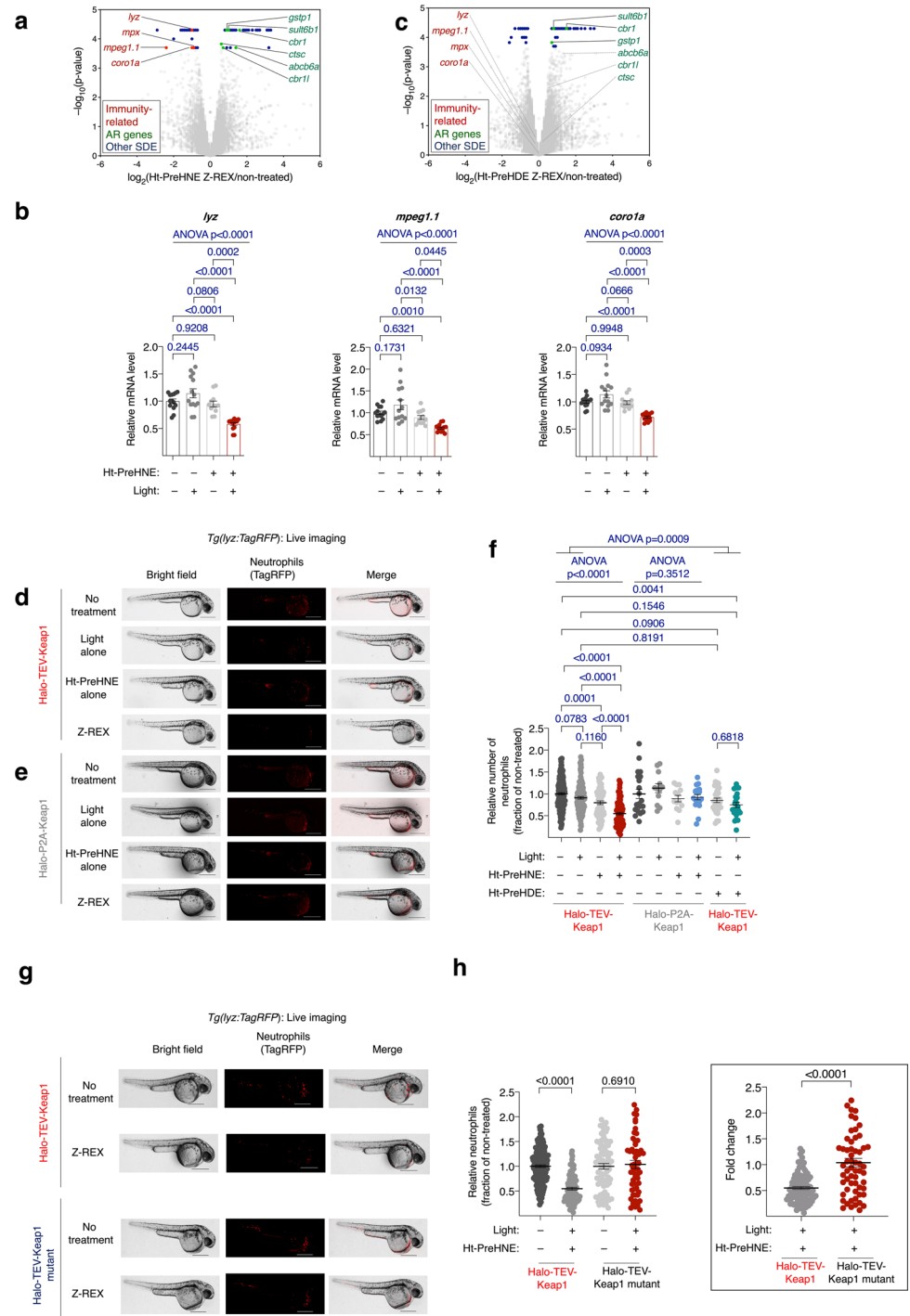

that Keap1-targeting, and expected Keap1-modification-promoted Nrf2-transcription factor-activation[19], had occurred.

Interestingly, several immune-cell-specific transcripts: *lyz*, *mpx*, *mpeg1.1*, and *coro1a* (Fig. 1a, in red; Supplementary Table 1, Supplementary Data 1), were depleted following Keap1-hydroxynonenylation. We substantiated this important result by qRT-PCR analysis integrating all possible Z-REX technical controls[14] (Fig. 1b, Supplementary Fig. 1a bottom row and 1b). Intriguingly, replicating the Z-REX–RNA-seq screen delivering HDE showed no depletion of immune-cell-specific transcripts (Fig. 1c; Supplementary Table 1, Supplementary Data 1), although similar upregulation of Nrf2-driven transcripts [Fig. 1c,

in green] was observed as expected, as with HNE. Critically, these data indicate that antioxidant response (AR) and immune-cell regulation through Keap1-electrophile engagement are uncoupled. We next progressed to: (1) examine the mechanisms for this chemo-specific behavior, and (2) examine its relevance to the mode-of action of DMF based on the mechanistic understanding gleaned in the first step.

**Immune-cell depletion accompanies Keap1-hydroxynonenylation.** We proposed two possible mechanisms that can explain the observed drop in immune cell-specific transcripts following

**Fig. 1 Immunity-related genes and neutrophils are suppressed in zebrafish embryos following Keap1-specific hydroxynonenylation. a** Differential expression from RNA sequencing of embryos subjected to Keap1-hydroxynonenylation by Z-REX. Statistically-significant differentially-expressed (SDE) Nrf2-driven AR genes marked with green dots; SDE immunity-related genes with red dots; all other SDE genes with blue dots; and non-SDE genes with gray dots. **b** qRT-PCR analysis validated suppression of immunity-related genes in zebrafish embryos following Z-REX-assisted Keap1-specific hydroxynonenylation. *P* values were calculated with ANOVA and Tukey's multiple comparisons test. **c** Same as in (**a**) except Z-REX-mediated modification of Keap1 was executed with a different electrophile, HDE, of similar Keap1-modification efficiency[53] and ability to upregulate AR genes [green dots]. Gray dashed lines mark genes not SDE in this comparison. See also Supplementary Table 1 and Supplementary Data 1. **d** Z-REX-mediated Keap1-hydroxynonenylation, but not Z-REX-technical controls (Supplementary Fig. 1a), caused depletion of neutrophil count in *Tg(lyz:TagRFP)*, in which neutrophils are labeled with TagRFP. Scale bars, 500 μm. **e** Same as in (**d**) in fish expressing Halo-P2A-Keap1, which cannot undergo Keap1-hydroxynonenylation by Z-REX (Supplementary Fig. 1a). Scale bars, 500 μm. **f** Quantitation of neutrophil levels in *Tg(lyz:TagRFP)* following Z-REX (against all technical controls) using photocaged Z-REX probes, Ht-PreH(D)NE, delivering HN(D)E. Note: signal-to-noise in these experiments (**d**–**f**) is 6:1, so gene expression changes rendering signal below detection levels cannot explain this loss of neutrophils. Tukey's multiple comparisons test was used to calculate corrected *p* values. **g** Similar experiment as in (**d**) in fish expressing either Halo-TEV-Keap1 or Halo-TEV-Keap1^C151S&C273W&C288E ('Halo-TEV-Keap1 mutant') which cannot undergo Keap1-hydroxynonenylation. Scale bars, 500 μm. **h** Quantitation of neutrophil levels in *Tg(lyz:TagRFP)* following Z-REX using photocaged Z-REX probes, Ht-PreHNE, delivering HNE (**d** and **g**). Inset: analysis of fold change (Z-REX/non-treated; see Fig. 2d) in neutrophil count. Note: for Halo-TEV-Keap1 set, the same data are presented in panels (**f**) and (**h**) for clarity. *P* values were calculated with two-tailed Student's *t*-test. All data present mean ± SEM. All *p* values for differential expression in RNA-seq were calculated with CuffDiff. All sample sizes are listed in Supplementary Methods. Source data are provided as a Source data file.

Keap1-specific hydroxynonenylation: a decrease in immune-cell counts or a suppression in immune-cell-specific gene expression, with no effect on immune cell number. A zebrafish line in which neutrophils are labeled with RFP [*Tg(lyz:TagRFP)*][20] was used to distinguish between these two possibilities. A mock Z-REX procedure had no effect on neutrophil counts (Supplementary Figs. 1a, 1c, 2a, b). By contrast, the total number of neutrophils present in live fish following Keap1-specific hydroxynonenylation was significantly reduced compared to all technical Z-REX-controls (Fig. 1d–f). These data explain the observed global mRNA depletion of neutrophil-specific genes.

For accurate interpretation, the experiments were replicated in embryos expressing a functional Keap1 unable to undergo Z-REX-proximity-assisted hydroxynonenylation (see "non-targetable P2A-construct", Supplementary Fig. 1a bottom row). No loss of neutrophils was observed (Fig. 1e, f). However, this control does not necessarily exclude off-target labeling of a protein complexed to Keap1. We thus performed Z-REX in fish expressing Halo-Keap1^C151S&C273W&C288E. This Keap1^C151S&C273W&C288E mutant retains Keap1^wt-protein's ability to bind target proteins such as Nrf2, but is inert to electrophile treatment[21]. Expression of Halo-Keap1^wt and Halo-Keap1^C151S&C273W&C288E were first validated to be similar (Supplementary Fig. 2c). However, no change in neutrophil levels was observed when Z-REX was replicated in larval fish expressing Halo-Keap1^C151S&C273W&C288E (Fig. 1g, h). These results together leave little doubt that neutrophil depletion is a Keap1-hydroxynonenylation-specific event, and that it is not due to adventitious hydroxynonenylation of other proteins. Such insights are not accessible using traditional genetic methods or other existing chemical biology/pharmacological approaches, as they cannot generate the modified state specifically, nor do they compare outputs to such precision negative controls.

Neutrophil depletion was transient: neutrophil counts were depleted 4 h post Keap1-specific hydroxynonenylation, but no significant drop was noted by 18 h (Supplementary Fig. 2d). Finally, consistent with our Z-REX–RNA-seq data (Fig. 1c), Keap1-specific engagement with HDE did not affect neutrophil count (Fig. 1f). As the chemical approaches used in HDE and HNE delivery are identical, this is clear evidence that Keap1 engagement by HNE, but not HDE, can lead to loss of neutrophils. This observation also rules out the possibility that the observed neutrophil loss is due to processes other than electrophile adduction that could, at least in principle, occur during Z-REX. A similar conclusion is also achieved from the experiments above that deployed the P2A construct (Fig. 1f,

Supplementary Fig. 1a bottom row) and Keap1 mutant (Fig. 1g, h, Supplementary Fig. 2c).

Further consistent with the original RNA-seq data, using an established macrophage-specific reporter strain *Tg(mpeg1:eGFP)*[22] (Supplementary Fig. 3a), a similar decrease in number of macrophages was observed following Keap1-specific hydroxynonenylation. Importantly, when zebrafish expressing both *mpeg1:eGFP* and *lyz:TagRFP* were analyzed, the reporter constructs gave rise to essentially no overlap, consistent with these reporters being expressed in different cell types (Supplementary Fig. 3b). The fold-drop in the neutrophil and macrophage counts was broadly similar to that measured in qRT-PCR analysis of immune-cell-specific mRNA abundance (Supplementary Fig. 3c). We thus conclude that the majority of the loss in immune-cell-specific transcript abundance derives from depletion of neutrophils/ macrophages as a consequence of Keap1-hydroxynonenylation in otherwise healthy developing embryos.

**Keap1-modification elicits apoptosis of neutrophils and macrophages.** We reasoned that depletion of neutrophils/macrophages could occur through either suppression of proliferation/ differentiation, and/or induction of death in these cells. To discriminate between these alternatives, we first tracked changes in neutrophil number following Keap1-hydroxynonenylation. In control embryos, over our typical time course, 4 h, the number of neutrophils increased by 1.5- to 1.8-fold (Fig. 2a, blue dots). In the same embryos post Keap1-hydroxynonenylation, there was no change in neutrophils over this period (Fig. 2a, red squares). Macrophage counts remained constant in control embryos over the same time course (Supplementary Fig. 4a, blue dots), but macrophage counts decreased following Keap1-hydroxynonenylation at early time points, returning to normal by 4 h (Supplementary Fig. 4a, red squares). When identical experiments were performed in embryos where HNE delivery to Keap1 was not possible during Z-REX (i.e., the P2A-system, Supplementary Fig. 1a bottom row), no change in neutrophil/macrophage number was observed relative to controls (blue dots vs. red squares in Fig. 2b, Supplementary Fig. 4b). We thus hypothesized that apoptosis was the likely cause of neutrophil/macrophage depletion. Consistent with this proposition, we detected colocalization of active Caspase-3 with neutrophils (red fluorescence) following Keap1-hydroxynonenylation (Fig. 2c, Supplementary Fig. 4c). Consistent with Keap1-hydroxynonenylation exerting little overall impact on fish development, there was no discernible upregulation of apoptosis across the rest of the embryo.

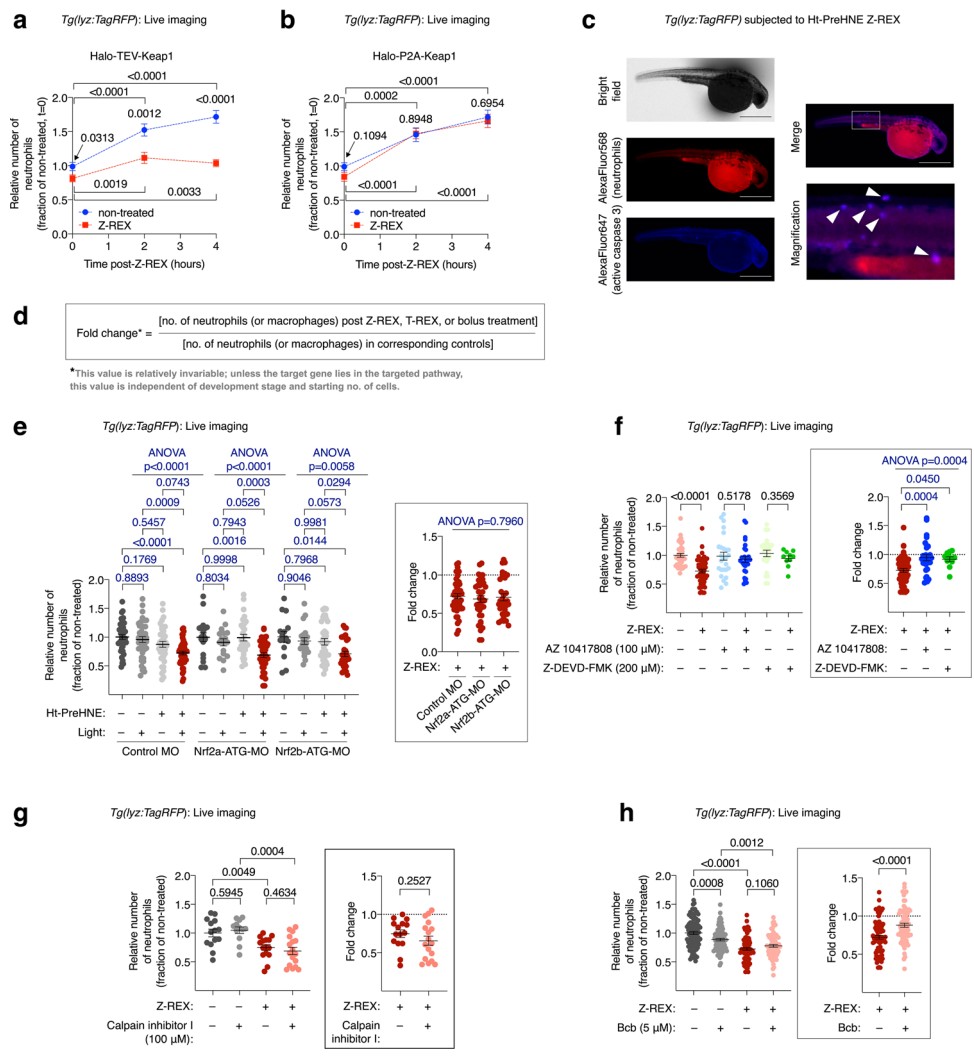

**Fig. 2 Keap1-hydroxynonenylation-promoted neutrophil loss proceeds via Nrf2-independent apoptosis. a, b** Neutrophil levels were tracked over time following Keap1-hydroxynonenylation by Z-REX (red squares) [against untreated control (blue dots)] in *Tg(lyz:TagRFP)* expressing either Halo-TEV-Keap1 (**a**) or Halo-P2A-Keap1 (**b**). Halo-P2A-Keap1 control construct (Supplementary Fig. 1a inset) cannot undergo Keap1-hydroxynonenylation via Z-REX. *P* values calculated with Student's two-tailed *t*-test. **c** Whole-mount immunofluorescence (IF) staining of active Caspase-3 (the key apoptotic executioner Caspase) and neutrophils in *Tg(lyz:TagRFP)* embryos following Keap1-hydroxynonenylation by Z-REX. Scale bars, 500 μm. Box in Merge panel marks the area magnified. White arrows mark colocalizations. **d** Equation used to calculate fold change (shown in inset panels). This value takes into account the starting number of immune cells and allows a fair comparison between different sets. The value is independent of starting immune-cell count and development. **e** Keap1-hydroxynonenylation with Z-REX was performed in *Tg(lyz:TagRFP)* embryos co-injected with control MOs, or MOs targeting Nrf2a or Nrf2b (500 μM). (ATG-MO: an MO targeting the translation start site; SPL-MO: an MO inhibiting splicing). Inset at right: analysis of fold change (Z-REX/non-treated) in neutrophil count. See Supplementary Fig. 5. *P* values were calculated with ANOVA and Tukey's multiple comparisons test. **f** Treatment of *Tg(lyz:TagRFP)* embryos with two inhibitors of Caspase-3 [AZ-10417808 (reversible, non-peptide) and Z-DEVD-FMK (covalent)] ablated Z-REX promoted neutrophil loss. Inset at right: analysis of fold change (Z-REX/non-treated) in neutrophil count. *P* values in black were calculated with two-tailed unpaired Student's *t*-test; *P* values in blue were calculated with ANOVA and Dunnett's multiple comparisons test. **g** Treatment of *Tg(lyz:TagRFP)* embryos with Calpain inhibitor I had no effect on Z-REX-promoted neutrophil loss. Inset at right: analysis of fold change (Z-REX/corresponding non-Z-REX-condition) in neutrophil count. *P* values were calculated with two-tailed unpaired Student's *t*-test. **h** Treatment of *Tg(lyz:TagRFP)* embryos with Bcb suppressed Z-REX-promoted neutrophil loss. Inset at right: analysis of fold change (Z-REX/corresponding non-Z-REX-condition) in neutrophil count. *P* values were calculated with two-tailed unpaired Student's *t*-test. All data present mean ± SEM. All sample sizes are listed in Supplementary Methods. Source data are provided as a Source data file.

**Keap1-modification-dependent apoptosis is Nrf2-independent.** Canonical Keap1/Nrf2/AR-signaling is essential for healthy development and also promotes resistance to chemical stresses[19]. This advantage extends to cancer cells, which often upregulate Nrf2 for survival[23]. Nonetheless, there are some sporadic links of Keap1/Nrf2/AR-signaling to apoptosis[24,25]. However, many

studies were performed using bolus-electrophile treatment[18] which is prone to artifacts and global toxicity. The Nrf2 pathway is widely postulated to be a target of DMF, although emerging data indicate that therapeutic efficacy of DMF is unaltered in Nrf2-knockout mice[26]. We investigated this (in)dependence by inducing Keap1-hydroxynonenylation in embryos where either

Nrf2a or Nrf2b (zebrafish-paralogs resulting from whole-genome duplication) was knocked down[27]. We observed no difference in suppression of immune-cell counts compared to control embryos (Fig. 2d, e, Supplementary Fig. 5a). These observations gave traction to our hypothesis that Keap1-specific-hydroxynonenylation mimics DMF-derived Keap1-succination, both of which result in loss of immune cells in a Nrf2-independent manner.

To substantiate these findings, we also examined the impact of Nrf2-overexpression or -knockdown alone on immune cells. Since Keap1-specific modification elicits Nrf2-upregulation, Nrf2-overexpression may be expected to promote neutrophil loss if Nrf2 were to mediate apoptosis, following Keap1-hydroxynonenylation. However, Nrf2-overexpression alone led to an increase in neutrophil counts (Supplementary Fig. 5b). Nrf2a-/Nrf2b-knockdown alone decreased neutrophil counts (Supplementary Fig. 5c). These data likely indicate that upregulation of Nrf2 or Nrf2-driven-genes [which our RNA-seq showed occurs post Keap1-modification (Fig. 1a, b)], is overall moderately antagonistic to immune cell loss.

**Immune-cell apoptosis requires Caspase-3 and Bax, but not Calpain**. To directly test whether immune cell loss was ascribable to apoptosis, we next screened the extent to which known pharmacological modulators of apoptosis affect Keap1-hydroxynonenylation-induced immune cell loss. Two different Caspase-3 inhibitors—Z-DEVD-FMK and AZ-10417808 (a highly selective, non-electrophilic, non-peptide-based caspase 3 inhibitor[28]) (Supplementary Fig. 2a)—ablated Keap1-hydroxynonenylation-promoted neutrophil- and macrophage depletion (Fig. 2d, f, Supplementary Fig. 6). These results indicate that immune-cell loss occurred through caspase-mediated apoptosis. Two principal arms of apoptosis feed into a common ultimate apoptosis effector, Caspase-3; namely: (i) Calpain, which signals through Caspase-12 and Caspase-9[29,30]; and (ii) mitochondrial-targeted apoptosis[31], that can function via Caspase-8[32] or Caspase-9[33] (Supplementary Fig. 6b). Although the relative importance of Caspases in these two pathways may be different, both pathways are dependent on Caspase to function effectively, especially hours post stimulus[34]. We found that a calpain inhibitor did not affect Keap1-hydroxynonenylation-promoted neutrophil depletion (Fig. 2d, g, Supplementary Fig. 6b).

We next examined the other arm of apoptosis (Supplementary Fig. 6b) using the Bax channel blocker, Bcb—an inhibitor of Bax-mediated mitochondrial-targeted apoptosis. We first showed that Bcb suppresses toxicity induced by the Bax-specific agonist SMBA1[35] in HEK293T cells (Supplementary Fig. 7a). These data also highlight that Bax-specific stimulation is sufficient to promote apoptosis[36,37]. Bcb-treatment of zebrafish embryos significantly suppressed Keap1-hydroxynonenylation-mediated loss of both neutrophils (Fig. 2d, h) and macrophages (Fig. 2d, Supplementary Fig. 7b). To examine the role of Bax in this apoptotic process further, we turned to Bax-knockout (KO) MEFs ($bax^{-/-}$) versus wild-type MEFs. We found that Bax KO MEFs were resistant to both bolus DMF and HNE treatment (Supplementary Fig. 7c), under conditions where the cells are able to proliferate during prolonged compound exposure. Under acute exposure to HNE or DMF, Bax KO suppressed PARP cleavage. Intriguingly, Bax KO was not sufficient to suppress PARP cleavage induced upon staurosporine treatment, even though Bax Bak Double-KO (where Bak is the other key member of the Bcl-2 family of pro-apoptotic proteins[38]) prevented PARP cleavage (Supplementary Fig. 7d). Thus, HNE and DMF share a similar pro-apoptotic mechanism that differs from that of staurosporine.

We thus returned to our experiments in zebrafish embryos, moving further upstream in the mitochondrial pathway (Supplementary Fig. 6b). We investigated inhibitors of p90RSK, an anti-apoptotic protein (chosen, in part, because it was previously implicated in DMF's mode-of-action, although no proof of necessity was furnished[15]). P90RSK promotes the action of Bcl-XL and Bcl-2[39], which restrict the pro-apoptotic protein Bax. p90RSK inhibitor (BI-D1870, Supplementary Fig. 2a) augmented the extent of macrophage loss (Fig. 2d, Supplementary Fig. 7e). Such a result could be rationalized by the fact that Keap1-hydroxynonenylation targets a pathway that functions similarly to p90RSK inhibition, i.e., a pathway negatively regulated by Bcl-XL/Bcl-2 (Supplementary Fig. 6b). A Bax-dependent mitochondrial apoptosis pathway is one clear possibility. These data thus add some extra credence to a Bax-promoted mechanism of apoptosis.

**Wdr1 links Keap1-specific hydroxynonenylation to apoptosis**. Having gleaned several independent lines of evidence for a Keap1-modification-specific mitochondrial apoptotic pathway dependent on Bax and Caspases (Supplementary Fig. 6b), we next examined how the interactome of Keap1 is altered in response to Keap1-hydroxynonenylation. These experiments were performed in a bid to link apoptosis upregulation with Keap1-specific hydroxynonenylation. We performed Keap1-pulldown experiments in SILAC[40]-HEK293T cells subjected to T-REX (which enables target-protein-specific hydroxynonenylation in cells[13]), against control (Supplementary Fig. 1a bottom row, Supplementary Fig. 8a), to identify proteins dissociating/associating with Keap1 following hydroxynonenylation. We identified 25 proteins that consistently, over 2 separate independent co-immunoprecipitation experiments, displayed altered affinity to Keap1 following Keap1-hydroxynoneylation: 11 proteins showed loss of Keap1 affinity and 14 showed a gain in interaction (Supplementary Fig. 8b–c; Supplementary Table 2; Supplementary Data 2). Among the proteins that showed no change in affinity to Keap1 was p62, a known Keap1 binder. All potential hits featuring zebrafish orthologs were carried forward to a knockdown-screen in zebrafish embryos. In only one instance (Zbed3) was no ortholog known in zebrafish. In several instances, the zebrafish ortholog had two paralogs. Thus, our final screening set in fish comprised 30 different morpholinos (MOs), targeting 24 different orthologs. Our validation screen was performed by evaluating Keap1-specific-hydroxynoneylation-dependent loss of neutrophils in embryos subjected to control MO vs. those subjected to MOs targeting specific genes. Any morphant showing significant rescue of Keap1-specific-hydroxynoneylation-dependent neutrophil loss was scored as a hit with potential functional relevance. MOs targeting *lmna*, *pou3f2a*, and *taf9* caused excessive toxicity and were eliminated from the screen. The pooled data showed that knockdown of most of our identified Keap1-binding proteins had little effect on neutrophil loss in fish (Fig. 3a, Supplementary Fig. 9a, b). Indeed, the average loss in neutrophils upon Z-REX for the whole set of 27 knockdowns and control was similar to neutrophil loss found in controls and embryos not exposed to any MO (Fig. 3a, central dashed line).

Intriguingly, two morphants showed no significant change in neutrophil counts upon Keap1-hydroxynonenylation. When the individual fold changes in neutrophil counts post Keap1-hydroxynonenylation were analyzed as an ensemble, these two morphants emerged to be greater than three standard deviations from the mean change in neutrophil counts of all the samples tested (Fig. 3a upper dotted line, Supplementary Fig. 9b); the corresponding candidate genes (neither of which are linked to DMF mechanism-of-action) were *slc4a1ap* and *wdr1*. Although

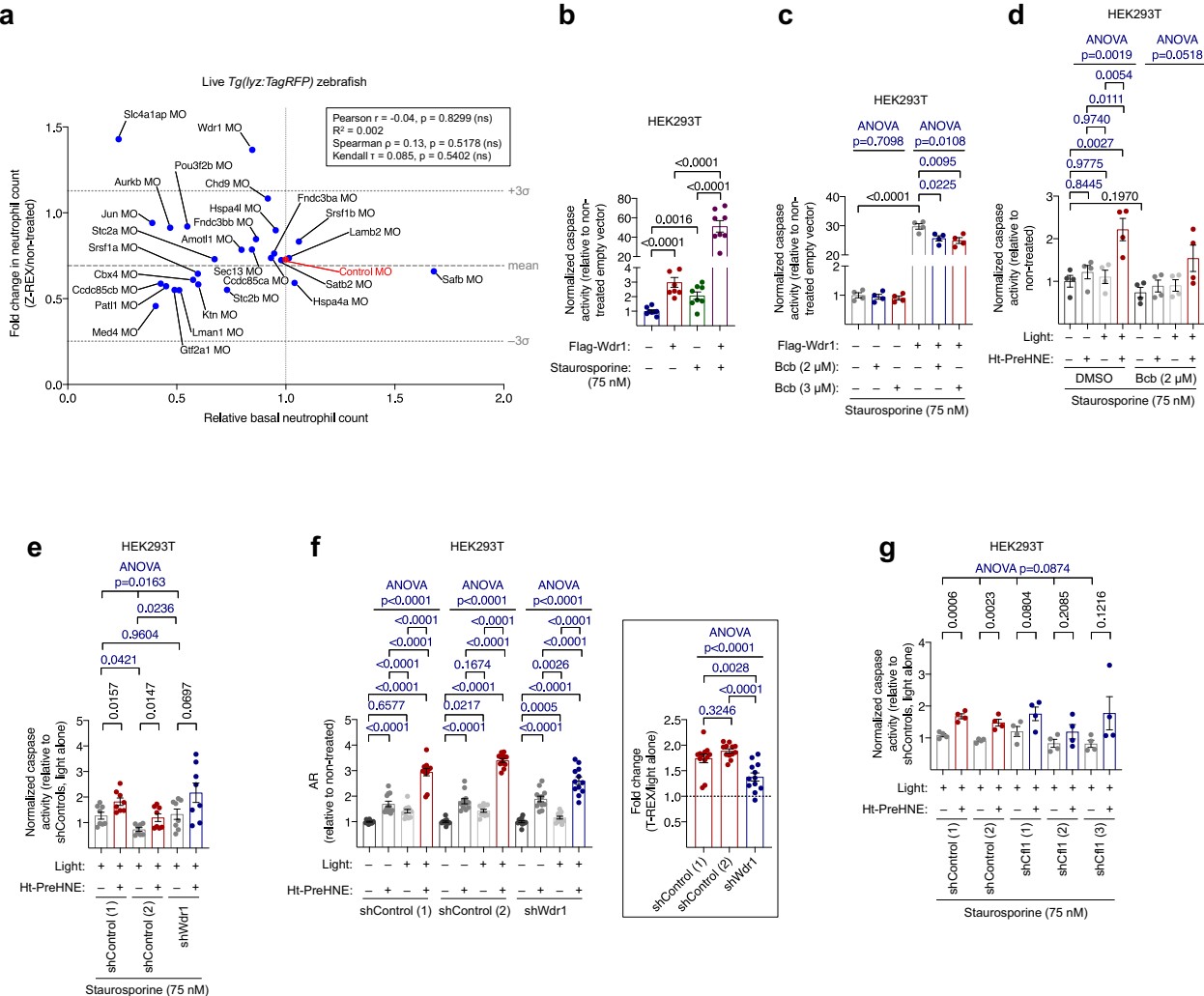

**Fig. 3 Wdr1 links hydroxynonenylated Keap1 to Bax-dependent apoptosis. a** Keap1-binding partners altered upon Keap1-specific-hydroxynonenylation identified by SILAC proteomics (see Supplementary Table 2, Supplementary Data 2) were knocked down using MOs targeting indicated genes in *Tg(lyz:TagRFP)*, and Z-REX-mediated Keap1-hydroxynonenylation was performed in these knockdown backgrounds. *X*-axis: effect of the knockdown alone; *Y*-axis: the fold change in neutrophil count following Z-REX-mediated Keap1-hydroxynonenylation. These effects were not correlated. See Supplementary Fig. 9a, b. **b** HEK293T transfected with either empty vector or Flag-Wdr1 were treated with staurosporine for 18 h and Caspase-3 activity was measured. *P* values were calculated with Student's *t*-test. **c** HEK293T transfected with either empty vector or Flag-Wdr1 were treated with staurosporine and/or Bcb for 18 h and Caspase-3 activity was measured. See Supplementary Fig. 11e, f. *P* values in black were calculated with Student's *t*-test; *P* values in blue were calculated with ANOVA and Dunnett's multiple comparisons test. **d** HEK293T subjected to T-REX-mediated Keap1-hydroxynonenylation and staurosporine stimulation showed Caspase-3-activity upregulation; treatment with Bcb ablated this effect. See Supplementary Fig. 11g. *P* values in black were calculated with Student's *t*-test; *P* values in blue were calculated with ANOVA and Tukey's multiple comparisons test. **e** HEK293T cells expressing shWdr1 or shControls were subjected to T-REX-mediated Keap1-hydroxynonenylation in the presence of staurosporine, and Caspase-3 activity was measured. See also Supplementary Fig. 12. *P* values in black were calculated with Student's *t*-test; *P* values in blue were calculated with ANOVA and Tukey's multiple comparisons test. **f** Antioxidant-response (AR) activity was measured in lysates from (**e**) to evaluate the intersection of Wdr1 with Keap1/Nrf2 signaling. Inset at right: Fold change (T-REX/light alone) in AR. *P* values were calculated with ANOVA and Tukey's multiple comparisons test. **g** Similar experiment as in (**e**) to measure Caspase-3 activity in HEK293T cells expressing shCfl1 or shControls. See Supplementary Figs. 12 and 15. *P* values in black were calculated with Student's *t*-test; *P* values in blue were calculated with ANOVA (see also Supplementary Fig. 1b). All data present mean ± SEM. All Student's *t*-tests were two-tailed and unpaired. All sample sizes are listed in Supplementary Methods. Source data are provided as a Source data file.

there was no correlation between change in neutrophils and basal neutrophil counts (by 3 different correlation tests), we noted that only Wdr1-knockdown had little effect on basal levels (i.e., levels under no Keap1-hydroxynonenylation) of neutrophils (see *x*-axis in Fig. 3a). Slc4a1ap-knockdown elicited a significant decrease in basal neutrophil counts. Further, in both runs of our independent cell-based SILAC data sets, the drop in the amount of Wdr1 bound to Keap1 following Keap1-specific-hydroxynonenylation, was one of the largest fold changes among all the hits [Supplementary Table 2 (entry 1) and Supplementary Data 2];

the fold change for Slc4a1ap was more subtle [Supplementary Table 2 (entry 19) and Supplementary Data 2]. We note that increased Nrf2-expression upregulated neutrophil numbers (Supplementary Fig. 5b). As discussed above, Nrf2-upregulation upon Keap1-hydroxynonenylation likely weakly countermands Keap1-hydroxynonenylation-promoted neutrophil apoptosis. We thus expected that when the apoptosis pathway is blocked, the neutrophil count may rise marginally, and this appeared to occur in Wdr1-morphants, albeit with $p = 0.068$ (Supplementary Fig. 9a), either way explaining the fact that there appears to be

neutrophil upregulation upon Keap1-hydroxynonenylation in Wdr1-morphants (Supplementary Fig. 9a, b).

Wdr1 loss-of-function in mice promotes initiation of incorrect inflammatory response by neutrophils and megakaryocytes; this process has been linked to incorrect apoptosis regulation particular to these cells[41]. We thus focused on Wdr1 in further studies. The association of overexpressed Keap1 and endogenous Wdr1, and the decrease in this interaction following HNE treatment, were validated in HEK293T cells by co-immunoprecipitation/western blot analysis (Supplementary Fig. 9c). Furthermore, endogenous Wdr1 emerged to be enriched upon IP of endogenous Keap1 (Supplementary Fig. 10). This association diminished when cells were treated with HNE (Supplementary Fig. 10).

**Hydroxynonenal targets Keap1/Wdr1-axis in macrophages and fish**. To glean mechanistic insights into this pathway and probe the pathway's role in apoptotic regulation in mammalian cells, we returned to cultured human cells where we had shown that Keap1-hydroxynonenylation releases Wdr1 from Keap1 binding (Supplementary Figs. 8 and 9c; Supplementary Table 2; Supplementary Data 2). We further found in HEK293T that Keap1-hydroxynonenylation upregulated endogenous Wdr1 expression (Supplementary Fig. 11a–c). Both Wdr1 release from Keap1, and upregulation in Wdr1 levels are reminiscent of how Keap1-hydroxynonenylation upregulates Nrf2/AR. We thus predicted that Wdr1 release from Keap1 actively upregulates apoptosis. Hence, apoptosis should be upregulated by Wdr1-overexpression. We observed enhanced Caspase activity in HEK293T cells ectopically expressing human Wdr1, irrespective of apoptosis stimulation by staurosporine (Fig. 3b). Consistent with our data above, Wdr1-induced apoptosis was, in this instance, partly dependent on Bax (Fig. 3c). Thus, increase in Wdr1 is sufficient to upregulate apoptosis through a Bax-dependent pathway (Supplementary Fig. 7b–d).

Returning to Keap1-hydroxynonenylation-specific apoptosis in HEK293T cells, we found this process was also strongly inhibited by a Caspase-3 inhibitor (Supplementary Fig. 11d). To examine how this electrophile-regulated Wdr1-dependent apoptosis interplays with the classical Keap1/Nrf2/AR axis, we measured AR-levels in cells overexpressing Wdr1 using a standard dual-luciferase reporter assay. A moderate increase in AR that was not dependent on Bax was observed (Supplementary Fig. 11e–g), agreeing with a previous report that indicates that Wdr1 may compete with Nrf2 for Keap1 binding[42]. Despite being modest in HEK293T cells (Fig. 3d, first 4-bars), we reasoned that Keap1-hydroxynonenylation-induced apoptosis upregulation likely functions through the same mechanism as occurs in neutrophils and macrophages, only on a smaller scale. We thus examined in HEK293T the dependence of Keap1-hydroxynonenylation-induced apoptosis on Bax. Consistent with our investigations in live zebrafish, in Bax KO MEFs, and the Wdr1-overexpression data in HEK293T, Bax-inhibitor suppressed Keap1-hydroxynonenylation-driven apoptosis in HEK293T cells (Fig. 3d); by contrast, Keap1-hydroxynonenylation-promoted AR-upregulation was unaffected by Bax inhibition (Supplementary Fig. 11g); thus, Keap1-modification-dependent pathways other than apoptosis are unaltered in cells treated with Bax-inhibitor Bcb. We also investigated the dependence of this apoptotic output on Wdr1. Consistent with results in embryos, Wdr1-knockdown HEK293T cells did not significantly upregulate caspase activity in cells, even though two control-knockdown lines were able upregulate caspase under the same conditions (Fig. 3e, Supplementary Fig. 12a, b). On the other hand, Wdr1-knockdown cells upregulated AR to a significantly lower extent than control cells (Figs. 2d and 3f), consistent with an intersection between Keap1/

Nrf2/AR and Keap1/Wdr1-apoptosis axes through Keap1. To reinforce this point, we examined how Wdr1 deficiency affected primary mouse bone marrow derived macrophages' (BMDMs) susceptibility to HNE. BMDM differentiation was validated using flow cytometry (Supplementary Fig. 13). Relative to two different shControl lines, two different corresponding shWdr1 lines [showing significant knockdown of Wdr1 (Supplementary Fig. 14a)] were resistant to HNE-induced PARP cleavage (Fig. 4a). Critically, cell-cycle analysis showed that Wdr1-knockdown lines were not significantly growth impaired, relative to shControl lines (Supplementary Fig. 14b). Altogether, there is thus a similar apoptotic pathway (i.e., one mediated by a common series of proteins under the same conditions) operative in mammalian cells, including primary macrophage cells, as there is in zebrafish neutrophils and macrophages.

**Cofilin is downstream of Wdr1 in modified-Keap1-driven apoptosis**. Wdr1 coordinates with cofilin to promote apoptosis through the mitochondrial pathway[43]. Thus, cofilin-knockdown should prevent Keap1-hydroxynonenylation-induced apoptosis. In three different cofilin-knockdown HEK293T cell lines (Fig. 3g), there was no significant upregulation of apoptosis upon Keap1-hydroxynonenylation. These lines were of similar knockdown efficiencies (Supplementary Fig. 12a, c), and also showed no change in Wdr1 levels (Supplementary Fig. 12b). Two knockdown-control lines showed significant upregulation of apoptosis under the same conditions. Cofilin-knockdown lines showed no significant difference in AR-upregulation from control lines (Fig. 2d, Supplementary Fig. 15). Thus, changes in AR-signaling cannot explain these observations.

**Wdr1 and cofilin mediate electrophile-driven immune cell apoptosis**. At this juncture, we were ready to further establish the importance of the pathway uncovered by Z-REX in live fish, using bolus administration of the native compounds. These experiments are critical to evaluate the validity of the mechanistic insights offered by Z-REX, and the relevance of the insights so gleaned to DMF's mechanism. This is because these experiments occur in wild-type fish and use electrophile treatment directly, yet they are guided by data obtained using T(Z)-REX-assisted targeted hydroxynonenylation. We thus determined the importance of Keap1, Wdr1, and cofilin in immune-cell-specific phenotypic modulation following whole-animal electrophile-administration. At the same developmental stage at which Z-REX was executed, bulk treatment of zebrafish embryos with HNE or DMF (25 or 50 μM) led to significant neutrophil and macrophage depletion (Fig. 4b). HDE treatment, in stark contrast, led to no change in neutrophils up to 25 μM (concentrations where HNE and DMF caused loss of neutrophils) (Fig. 4b). Above 25 μM, HDE was toxic. These data are entirely consistent with HNE mimicking DMF, but HDE failing to do so, as predicted by Z-REX.

The loss of neutrophils caused by whole-animal DMF- or HNE-administration was Nrf2-independent (Fig. 2d, Supplementary Fig. 16), as also predicted by our Z-REX data (Fig. 2e). We next evaluated if the players we discovered above were involved in immune-cell-loss promoted by bulk-electrophile-exposure. Bax inhibition completely suppressed the loss of neutrophils when embryos were treated with either HNE or DMF (Figs. 2d, 4c). We then investigated proteins upstream of Bax. Remarkably, two different MOs targeting Wdr1 led to total suppression of bolus-HNE- or -DMF-promoted neutrophil/macrophage loss (Fig. 5a, Supplementary Fig. 17a, b). To further examine the role of Wdr1 in DMF-induced neutrophil apoptosis, we took advantage of fish

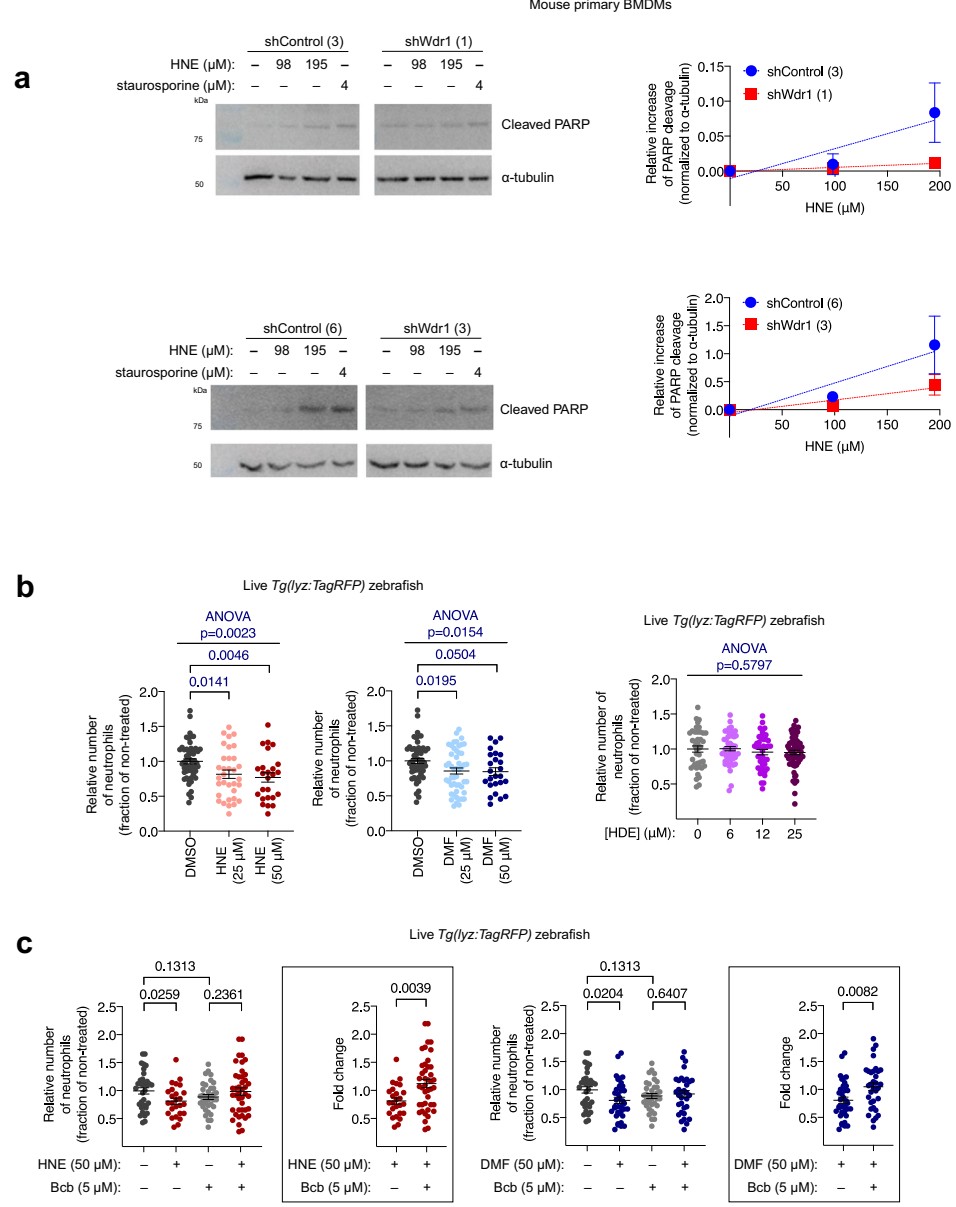

**Fig. 4 Bulk exposure of mouse primary BMDMs and larval zebrafish to HNE/DMF triggers Bax-dependent macrophage/neutrophil apoptosis. a** Left: western blots analyzing changes in the endogenous levels of cleaved PARP (hallmark of apoptosis) in two different pairs of lentiviral shWdr1-knockdown BMDMs and knockdown-control BMDMs, subsequent to HNE or staurosporine treatment (3 h). Right: ImageJ quantification of relative extent of cleaved PARP in HNE-treated, compared to the respective DMSO-control set, in Wdr1-knockdown BMDMs (red squares) vs. shControl (blue dots) lines. In each dataset, normalization was performed against α-tubulin loading control. Slopes from linear regression of shControl vs. shWdr1, Upper plot: 0.00043 ± 0.00012 and 0.00006 ± 0.00003, respectively. Lower plot: 0.0059 ± 0.0022 and 0.0023 ± 0.0008, respectively. Fits derived from fitting to $y = mx$. **b** Tg(lyz:TagRFP) embryos expressing Halo-TEV-Keap1 were treated at 30 hpf with HNE or DMF or HDE for 6 h at indicated concentrations (up to maximum tolerable amount prior to cytotoxicity), and then neutrophils were counted. P values were calculated with ANOVA and Dunnett's multiple comparisons test. Note that for the left two plots, the non-treated data are the same, but are presented in each plot for clarity. **c** Similar experiment as in (**a**) except with or without simultaneous administration of Bcb. Inset at right: Fold changes (treated/non-treated) in neutrophil counts. P values were calculated with two-tailed Student's unpaired t-test. Note that the non-treated data are the same, but are presented in each plot for clarity. All data present mean ± SEM. All sample sizes are listed in Supplementary Methods. Source data are provided as a Source data file.

carrying the *Carmin* allele, encoding a defunct Wdr1 protein[44]. Early embryos with either one or two *Carmin* alleles were resistant to DMF treatment (Fig. 5b, Supplementary Fig. 18a–d). Our data further predict that cofilin-knockdown should also suppress HNE- and DMF-induced neutrophil apoptosis. This was in fact the case for two separate cofilin-knockdowns (Fig. 5a, Supplementary Fig. 17c).

Most importantly, knockdown of Keap1a/b also fully suppressed DMF-induced neutrophil loss (Fig. 5c, Supplementary Fig. 19). We compared these results head to head with fish in which GAPDH, a key alternative target of DMF[10], had been knocked down. Under neither Z-REX nor bolus-HNE/DMF-dosing conditions did GAPDH-knockdown suppress neutrophil loss (Supplementary Fig. 20a–c).

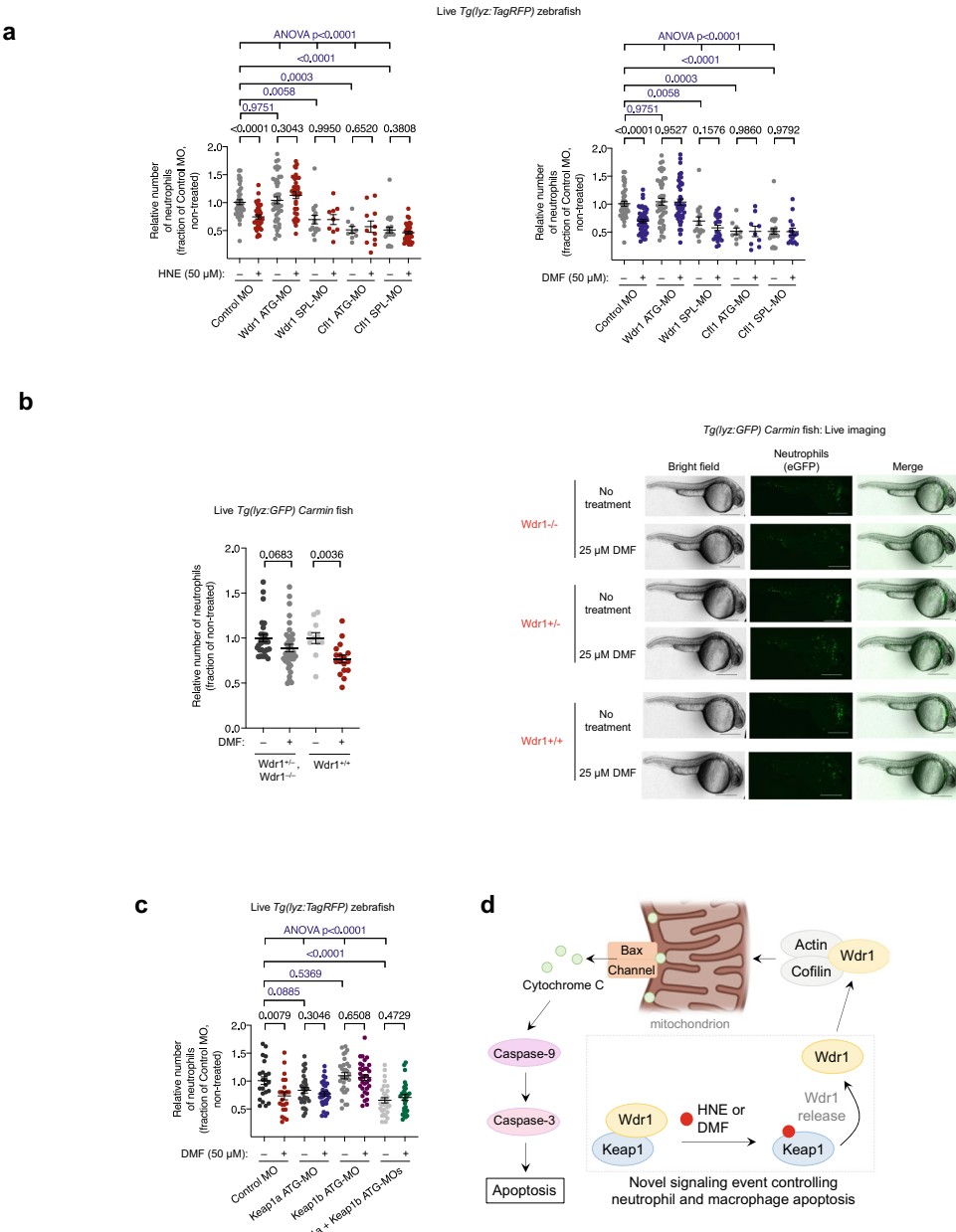

**Fig. 5 Bulk exposure of embryos to HNE or DMF triggers Wdr1- and cofilin-dependent apoptosis of neutrophils. a** Similar experiment as in Fig. 4b except *Tg(lyz:TagRFP)* embryos were injected with Halo-TEV-Keap1 and the indicated MO (500 μM). See also Supplementary Fig. 16. In all cases, embryos were exposed to HNE or DMF at 30 hpf, and neutrophils were counted at 36 hpf. Inset at right: Fold changes (treated/non-treated) in neutrophil counts in each knockdown background. *P* values in black were calculated with two-tailed unpaired Student's *t*-test. *P* values in blue were calculated with ANOVA and Dunnett's multiple comparisons test (see also Supplementary Fig. 1b). Note that the non-treated data sets in each plot are the same, but presented in each plot for clarity. **b** *Tg(lyz:eGFP) carmin* embryos were treated at 30 hpf with DMF for 4 h, and neutrophils were counted subsequently. See also Supplementary Fig. 18a–d. Scale bars, 500 μm. Inset: analysis of fold change (DMF/non-treated) (see Fig. 2d) in neutrophil count. *P* values were calculated with two-tailed unpaired Student's *t*-test (Supplementary Fig. 1b). **c** g(lyz:TagRFP) embryos were injected with MOs targeting Keap1a, Keap1b, both, or control MO (500 μM). See also Supplementary Fig. 19. At 30 hpf, embryos were treated with DMF for 6 h, then neutrophils were counted. *P* values in black were calculated with two-tailed unpaired Student's *t*-test. *P* values in blue were calculated with ANOVA and Dunnett's multiple comparisons test (see also Supplementary Fig. 1b). **d** Model of the Keap1-modification-specific apoptotic axis necessary for DMF mode-of-action in immune-cell depletion. Upon Keap1-electrophile-modification, Wdr1 is released from Keap1 binding. In concert with cofilin, Wdr1 promotes Bax-dependent mitochondrial-targeted activation of Caspase-3. All data present mean ± SEM. All sample sizes are listed in Supplementary Methods. Source data are provided as a Source data file.

## Discussion

These data present evidence that Keap1 is a sufficient target of the drug Tecfidera® (DMF), able to quantitatively account for the gamut of the drug's apoptotic effects on macrophages and neutrophils. Such effects are strongly associated with drug action and have been observed in patients. Critically, we show that a specific target of this highly-electrophilic drug is a biologically sufficient responder under several conditions: not only conditions of precision electrophile–target engagement, relative to precise controls (i.e. using T(Z)-REX), but also when the target—and, separately, specific pathway components—are knocked down/out during bolus electrophile/electrophilic drug exposure. In the latter

case, our specific phenotype is apoptosis in innate immune cells of an otherwise healthy developing animal or primary cells. Among several immune cell types, we focused on neutrophils and macrophages as representative effectors of chronic inflammation and demyelination in relapsing multiple sclerosis[45] for which DMF treatment proves successful. In these relevant cell types, our data demonstrate that Keap1 is an intrinsically important target that is sufficient to recapitulate clinical behaviors. How manifestations of apoptosis in each different immune cell funnels into overall drug efficacy, or contributes to unwanted side-effects, is a remaining unknown that needs to be studied in subsequent work. However, it is encouraging that models of multiple sclerosis exist in zebrafish[46–49], and through integration of tools used in this paper and those model systems, more insights will likely be possible. Thus, our experimental approaches offer a unique means to specifically mimic drug action in specific cells and afford a testable model for further investigations.

As has been stated in all previous studies on DMF and analogs, these data cannot absolutely rule out other targets of the drug. However, in contrast to existing data[10–12,15], herein we compare outputs of both target-specific effects (wherein precision negative controls allow us to minimize the impact of off-target effects) and bolus electrophile/electrophilic drug dosing. This comparison shows each condition gives the same magnitude of outputs, with the same obligatory regulators. Thus, this study quantitatively assigns the importance of the uncovered pathway (Fig. 5d), in otherwise healthy and unperturbed fish and cells. To give an example of the power of this approach, cofilin, one of the downstream players in our described pathway, has been proposed to be a target of DMF[50]. In the target-rich world of pleiotropic drugs, such quandaries are common. However, by combining mutagenesis, knockdown/knockout, and ramifications of precision on-target signaling, our data show that cofilin labeling by DMF is not needed for this pathway to function. Such analysis has not been possible previously: either proposed target knockdown has not affected susceptibility to DMF[51], or interrogation of on-target labeling has not been possible (aside from via the use of other electrophilic compounds[10], which have other off-target effects, likely including Keap1).

Of course, there are limitations to our strategy. Issues with the Z-REX method, such as delivery to unintended proteins can be dealt with through mutagenesis and other controls, such as the non-fused P2A construct we use here. However, we cede that such electrophile-sensing defective but otherwise functional mutants may not always be available. Perhaps more importantly, currently Z-REX is limited to enal/enone-derived natural electrophiles, in terms of the class of reactive molecules it can release. This limitation forced us to identify (a) suitable enal/enone-based natural electrophile(s) as a surrogate for DMF. Our Z-REX-coupled RNA-Seq data and associated downstream results surprisingly revealed that HNE—but not HDE, another natural electrophile related to HNE—is a good surrogate for DMF-promoted Keap1-Wdr1-dependent immune-cell apoptosis pathway that we discovered. However, HDE behaves similarly to HNE in terms of canonical Keap1-Nrf2-antioxidant-response signaling[13,52]. Interestingly, when we examined this phenomenon in several other known stress-responsive pathways using numerous luciferase reporter assays (Supplementary Fig. 21a, b), we found that this similarity in output is a more or less general property of DMF and HNE, versus HDE. Given the complexity of the systems, the underlying explanations for such observations under bulk-electrophile exposure could be manifold, and may not all be linked to Keap1. However, either way, these data overall do provide credence that DMF and HNE function similarly.

Perhaps the most interesting observation to emerge from this work is the heightened susceptibility of several immune cells to DMF-induced apoptosis. It is worth noting, as mentioned above, that this observation is consistent with clinical observations, and hence indicates that our model system recapitulates response to DMF well. Critically, Z-REX—as a quasi-intramolecular targeted delivery—allows regulation of the percentage of target protein modified in cells. Thus, these data show that it is likely not permeation/distribution of the electrophile into a subset of cell types that is responsible for selective toxicity/activity as has been seen with several other approved small-molecule drugs, e.g., bortezomib. Rather, it is a specific susceptibility of immune cells to the specific Keap1-orchestrated apoptosis pathway we outline here. As alluded to above, Wdr1 is indeed known to be particularly important for function of neutrophils in several model systems. Wdr1 is an essential gene, however, which has limited study somewhat. Nevertheless, Wdr1 mutant mice show autoinflammatory disease and changes in neutrophil behavior[53]. Mutations in Wdr1 also manifest themselves in human neutrophils as well[54]. Thus, neutrophils and macrophages, and likely other immune cells, are particularly sensitive to Wdr1 levels.

We thus propose that the combination of Z-REX and bolus dosing will be useful to parse molecular targets of electrophiles in clinical use or trials. Since we chose to use DMF as our test electrophilic drug, and HNE/HDE as our comparison electrophiles, this work also shows that a number of varied electrophiles may be able to be modeled through the use of T(Z)-REX. Of course, as the classes of electrophiles compatible with T(Z)-REX increase, the range of processes available to such studies will increase, which is also encouraging. In a broader sense, these data further document the complex regulatory modes present in biological systems. Namely, Keap1, a protein believed to be essential for survival, can turn tail and regulate apoptosis in certain cell types. Understanding of how this cell-type-specific mechanism functions is likely the next phase of these investigations.

## Methods

**Statistics and reproducibility.** Sample sizes and numbers of biological replicates are listed in Supplementary Methods. For zebrafish experiments, each larval fish was considered a biological replicate. When images of zerbafish are shown, they are representative of multiple independent samples (see Supplementary Methods). For experiments involving cultured cells, samples generated from individual wells or plates were considered biological replicates. In the figure legend for each experiment, how the data are presented in the figure (typically mean ± SEM) is clearly indicated. Our statistical approach is detailed in Supplementary Fig. 1b. Data were plotted/fit and statistics generated using GraphPad Prism 7 or 8.

**Reagents.** All HNE and HDE used in this study were alkyne-functionalized. These molecules are often referred to elsewhere as "HNE(alkyne)" and "HDE(alkyne)", respectively, but this convention is not used here for clarity. These reagents were synthesized as previously described[55]. Ht-PreHNE and Ht-PreHDE were synthesized as previously described[52]. Unless otherwise indicated, all other chemical reagents were bought from Sigma at the highest availability purity. DMF was from Alfa Aesar. TCEP was from ChemImpex. Staurosporine was from Apollo Scientific. Calpain inhibitor I was from Cayman Chemical. AZ-10417808, Z-DEVD-FMK, Z-IETD-FMK, Bax channel blocker, and BI-D1870 were all from ApexBio. Ac-DEVD-AMC was from Bachem. Puromycin was from Santa Cruz. SMBA1 was from Sigma. AlamarBlue was from Invitrogen, and was used according to the manufacturer's instructions. Minimal Essential Media, RPMI, Opti-MEM, Dulbecco's PBS, 100X pyruvate (100 mM), 100X nonessential amino acids (11140-050), and 100X penicillin–streptomycin (15140-122) were from Gibco. Isotopically pure amino acids were from Sigma. Protease inhibitor cocktail complete EDTA-free was from Roche. TALON (635503) resin was from Clontech. 2020 and LT1 transfection reagents were from Mirus. PEI was from Polysciences. Tricaine methanesulfonate was from Sigma. Venor GeM PCR-based mycoplasma detection kit was used as stated in the manual and was from Sigma. ECL substrate and ECL-Plus substrate were from Pierce and were used as directed. Acrylamide, ammonium persulfate, TMEDA, and Precision Plus protein standard were from Bio-Rad. All lysates were quantified using the Bio-Rad Protein Assay (Bio-Rad) relative to BSA as a standard (Bio-Rad). PCR was carried out using Phusion Hotstart II (Thermo Scientific) as per the manufacturer's protocol. All plasmid inserts were validated by sequencing at Cornell Biotechnology sequencing core facility or Microsynth. All sterile cell culture plasticware was from CellTreat. Sources of antibodies are listed in Supplementary Table 6. Antibodies were validated using western blot/

immunofluorescence to show that multiple independent shRNAs/MOs significantly reduced the levels of the detected protein. Sources of luciferase reporter plasmids are listed in Supplementary Table 8.

**Construction of plasmids**. Sequences of all primers used for cloning are listed in Supplementary Table 3. The inserts in all plasmids generated were fully validated by Sanger sequencing at the Cornell Genomics Core Facility or Microsynth. pCS2 + 8 Flag-Wdr1 was constructed by amplification of wdr1 (obtained from the EPFL Gene Expression Core Facility) using the appropriate forward and reverse primers with Phusion Hotstart II following the manufacturer's protocol. The PCR products were then extended using forward and reverse extension primers. Empty pCS2 + 8 plasmid was digested with EcoRI-HF (NEB), and the extended PCR product was cloned into this plasmid.

pCS2 + 8 Halo-TEV-Keap1$^{C151S,C273W,C288E}$ was constructed from pCS2 + 8 Halo-TEV-Keap1 plasmid, using the appropriate forward and reverse primers containing desired mutations with Phusion Hotstart II following the manufacturer's protocol. The PCR products were then extended using forward and reverse extension primers. pCS2 + 8 Halo-TEV-Keap1-2HA plasmid was digested with SphI-HF (NEB, R3182), and the extended PCR product was cloned into this plasmid.

**Zebrafish husbandry and breeding**. All procedures at Cornell University were approved by the Institutional Animal Care and Use Committee (IACUC) and performed in accordance with the guidelines of the NIH. All procedures at EPFL were performed in accordance with the Swiss regulations on animal experimentation (Animal Welfare Act SR 455 and Animal Welfare Ordinance SR 455.1), in the EPFL zebrafish unit, cantonal veterinary authorization VD-H23. Tg(lyz:TagRFP) were obtained from the Zebrafish International Resource Center (ZIRC, University of Oregon). Tg(mpeg1:eGFP) were obtained from Professor Todd Evans (Weill Cornell Medicine). These reporter lines were crossed with WT fish to produce progeny exclusively heterozygous for the fluorescent marker (or wt/wt, which were not examined). Embryos were produced by natural mating in breeding tanks.

Tg(lyz:eGFP) Carmin fish were obtained from Professor Philippe Herbomel (Institut Pasteur). Tail fins were clipped from adult fish and submerged in 30 μL lysis buffer (10 mM Tris, 1 mM EDTA, 0.3% Tween-20, and 0.3% NP-40). The mixture was heated to 98 °C for 10 min, cooled on ice for 1 min, supplemented with 3 μL of 20 mg/mL proteinase K (Sigma-Aldrich, P2308), and incubated at 55 °C for 14 h. The resulting genomic DNA solution was then used to PCR amplify the desired gene sequence (Supplementary Table 3). The PCR product was characterized by Sanger sequencing at Microsynth to identify adult fish genotype (sequencing primer: 5′-GCTTGATCCTCTTTCTCAGT-3′, Supplementary Table 3). Heterozygous carmin fish were crossed with carmin heterozygotes in the Tg(lyz:eGFP) background fish to produce progeny, whose genotypes were later identified by a similar genotyping method described above, but with whole fish embryos instead of fin clips. Genotypes of embryos were also characterized by restriction enzyme digestion of PCR products: the PCR product generated as described above was incubated with NciI (100 units/mL, R0196, NEB) in 1x cutsmart buffer (B7204, NEB) at 37 °C for 2 h, followed by agarose gel electrophoresis (Supplementary Fig. 18a–c).

**Zebrafish microinjection**. The yolk sack of embryos at the 1–4-cell stage was microinjected with ~2 nl of mRNA [1.4–1.6 mg/ml (mMessage mMachine SP6 kit, Invitrogen)] or 2 nl of morpholino oligonucleotides [0.5 mM (Supplementary Table 4; GeneTools LLC)] or 2 nl of a 1:1 mixture of mRNA and MO (0.5 mM) as indicated in the figures. Fish were grown at 28.5 °C with a 14-h light/10-h dark cycle in 10% Hank's Balanced Salt Solution (HBSS).

**Z-REX procedure in zebrafish**. This was carried out as previously reported[14]. Embryos injected with mRNA encoding Halo-TEV-Keap1-2xHA or Halo-2xHA-P2A-TEV-Keap1-2xHA as described above were treated with preHNE(alkyne) (1 μM) in 10% HBSS containing penicillin–streptomycin (Gibco) and grown in the dark for 30 h. Embryos were then rinsed with 10% HBSS three times for 30 m each in a red-light room, and exposed to UV light (365 nm, 0.5 mW/cm²) for 3 m with gentle agitation of the plate every minute. Embryos were then either immediately euthanized or incubated further as indicated. For Z-REX experiments with small-molecule treatment (e.g., Bcb), the compound was introduced at the stated concentration in the 10% HBSS immediately following light exposure.

**RNA sequencing of zebrafish embryos**. Zebrafish expressing Halo-TEV-Keap1-2xHA were subjected to Z-REX conditions. Euthanized embryos were lysed in Trizol using ceramic hard tissue homogenizing mix (VWR), and RNA was isolated following the manufacturer's protocol. The quality of the RNA was assessed by Nanodrop spectrophotometry (A260/A280 ratio ~2) and fragment analysis using a Bioanalyzer (Agilent). The rest of the procedure including data processing was performed by the RNA sequencing core (RSC) service at Cornell University. Briefly, from 500 ng total RNA input, polyadenylated RNA was isolated with the NEBNext Poly(A) mRNA Magnetic Isolation Module (New England Biolabs). TruSeq-barcoded RNA-seq libraries were generated with the NEBNext Ultra

Directional RNA Library Prep Kit (New England Biolabs). Each library was quantified with a Qubit 2.0 (dsDNA HS kit; ThermoFisher) and the size distribution was determined with a Fragment Analyzer (Advanced Analytical) prior to pooling. Libraries were sequenced on a NextSeq500 instrument (Illumina). At least 20 M single-end 81nt reads were generated per library. Reads were trimmed for low quality and adapter sequences with cutadapt v1.8 [parameters: –m 50 –q 20 –a AGATCGGAAGAGCACACGTCTGAACTCCAGTC –match-read-wildcards[56]]. Reads were mapped to the reference genome/transcriptome (D. rerio GRCz10 [Ensembl]) using tophat v2.1 [parameters: –library-type=fr-firststrand –no-novel-juncs –G<ref_genes.gtf>[57]]. Cufflinks v2.2 (cuffnorm/cuffdiff) was used to generate FPKM values and statistical analysis of differential gene expression[58].

**Quantitative real-time PCR (qRT-PCR)**. This was carried out as previously described[59]. Briefly, embryos treated as indicated (3–5 per sample) were euthanized, washed with PBS, and homogenized in Trizol reagent with glass beads and vortexing for ~30 s. Total RNA was then isolated following the manufacturer's protocol. One microgram of total RNA (purity/integrity assessed by agarose gel electrophoresis and concentration determined by A$_{260nm}$ using a BioTek Cytation3 microplate¯ reader with a Take3 accessory) was treated with amplification-grade DNase I (Invitrogen) and reverse-transcribed using Oligo(dT)$_{20}$ as a primer and Superscript III Reverse Transcriptase (Life Technologies) following the manufacturer's protocol. PCR was performed for two technical replicates per sample using iQ SYBR Green Supermix (Bio-Rad) and primers specific to the gene of interest (Supplementary Table 3) following the manufacturer's protocol. Amplicons were chosen that were 150–200 bp in length and had no predicted off-target binding predicted by NCBI Primer BLAST. For genes with multiple splice variants, primers were chosen that amplified conserved sequences across all splice variants. Primers were validated using standard curves generated by amplification of serially-diluted cDNA; primers with a standard curve slope between –0.8 and 1.2 and $R^2 \geq 0.97$ were considered efficient. Single PCR products were confirmed by melting analysis following the PCR protocol. Data were collected using a Light-Cycler 480 (Roche). Threshold cycles were determined using the LightCycler 480 software. Samples with a threshold cycle >35 or without a single, correct melting point were not included in data analysis. Normalization was carried out using a single housekeeping gene as indicated in each dataset and the $\Delta\Delta C_t$ method.

**Live imaging of macrophages and neutrophils in zebrafish embryos**. Following treatment as indicated, embryos were anesthetized with tricaine methanesulfonate (Sigma), manually dechorionated, and imaged on a 2% agarose pad using a Leica M205-FA fluorescence stereomicroscope. Fluorescent macrophages and neutrophils were quantitated using the "Find Maxima" function of ImageJ (NIH).

**Immunofluorescence (IF)**. Whole-mount IF was carried out as previously described[60]. Briefly, euthanized embryos washed twice with PBS were fixed in 4% formaldehyde (1 ml) overnight at 4 °C with gentle rocking in 1.5 ml tubes. Formaldehyde was replaced with chilled methanol (1 ml) and embryos were kept at –20 °C overnight. Methanol was replaced with PDT [1 ml; 1X PBST (PBS with 0.1% Tween-20), 1% DMSO, 0.3% Triton-X-100], which was replaced with fresh PDT, and embryos were gently rocked for 30 min at room temperature. The PDT was again replaced with fresh PDT and embryos were rocked for another 30 min. PDT was replaced with blocking buffer (1 ml; 1X PBST, 10% FBS, 2% BSA) and embryos were gently rocked for 1 h. Primary antibodies (Supplementary Table 6) were diluted to the appropriate concentration in fresh blocking buffer (0.8 ml). Embryos were gently rocked with antibodies for either 2 h at room temperature or overnight at 4 °C. The washing and blocking procedure was repeated, and embryos were gently rocked with secondary antibodies in blocking buffer (1 ml) for 1.5 h at room temperature. The washing (but not blocking) procedure was again repeated, and stained embryos were immediately imaged on a 2% agarose pad using a Leica M205-FA fluorescence stereomicroscope.

**Cell culture**. HEK293T (obtained from ATCC) and were cultured in MEM (Gibco 51090036) supplemented with 10% v/v fetal bovine serum (Sigma), penicillin/streptomycin (Gibco), sodium pyruvate (Gibco), and nonessential amino acids (Gibco) at 37 °C in a humidified atmosphere of 5% CO$_2$. Media were changed every 2–3 days.

MEFs [obtained from ATCC: CRL-2907 (wt), CRL-2910 ($bax^{-/-}$), and CRL-2913 ($bax^{-/-}bak^{-/-}$] were cultured in IMDM (Gibco 12440-053) supplemented with 10% v/v fetal bovine serum (Sigma), penicillin/streptomycin (Gibco), sodium pyruvate (Gibco), and nonessential amino acids (Gibco) at 37 °C in a humidified atmosphere of 5% CO$_2$. Media were changed every 2–3 days.

**Mouse maintenance and culture of bone marrow cells (BMCs)**. The mouse study protocols/experimental procedures underwent an ethical review and were approved by the Swiss Veterinary Authorities (license no. VD3290). Mice were maintained in standard housing with a temperature of 22 ± 2 °C, relative humidity of 55 ± 10%, and a 12 h:12 h light:dark cycle (7 am–7 pm). BMCs were extracted from mouse femurs and tibiae as previously described[61], and differentiated to macrophages (bone marrow derived macrophages, BMDMs) by adding 50 ng/mL

recombinant murine M-CSF (macrophage colony-stimulating factor, PeproTech, 315-02) to the growth medium as previously described[62]. BMCs were cultured in DMEM (Gibco 41965-039) supplemented with 10% v/v fetal bovine serum (Sigma), penicillin/streptomycin (Gibco), and sodium pyruvate (Gibco) at 37 °C in a humidified atmosphere of 5% $CO_2$. Media were changed every 2–3 days.

**General T-REX procedure in cells.** HEK293T cells were transfected as indicated. After 24 h, cells were treated with Ht-PreHNE (5 μM) for 2 h in serum-free media. Excess Ht-PreHNE was removed by replacing the media with fresh serum-free media three times over a period of 1.5 h. Cells were then exposed to light (365 nm, 0.5 mW/cm², 3 min) and incubated further as indicated. In cases where small-molecule (e.g., Bcb) treatment was necessary, compounds were introduced in serum-free media at the indicated concentrations immediately following light exposure.

**Analysis of Keap1 interactome changes following Keap1-hydroxynonenylation by T-REX-coupled SILAC proteomics strategy.** A previously-generated HEK293T line expressing DsRed-IRES-His₆-Halo-TEV-Keap1[59] was grown for six passages in media containing exclusively "light" lysine and arginine (K + 0, R + 0) or "heavy" lysine and arginine (K + 8, R + 10), with media changes every 2–3 days. These cells were cultured in duplicate 10 cm diameter plates. Cells were treated with Ht-preHNE(alkyne) (20 μM) for 2 h in serum-free media (with "light" or "heavy" amino acids as appropriate). Cells were then rinsed three times with serum-free media (30 m per rinse) and exposed to UV light (365 nm, 0.5 mW/cm²) for 5 m. Cells were incubated for a further 1.5 h, then harvested, counted using a Countess II FL cell counter, and heavy and light cells were mixed in a 1:1 ratio for each comparison. Pooled cells were then lysed in lysis buffer [50 mM HEPES pH 7.6, 150 mM NaCl, 1% NP-40, 5 mM imidazole, and 1X cOmplete EDTA-free protease inhibitor tablets (Roche)] using three rapid cycles of freeze–thaw–vortexing with glass beads (Sigma). Insoluble material was cleared by centrifugation and total protein was diluted to 1 mg/ml and incubated with 50 μl TALON resin (Takara Bio) for 1 h at 4 °C with end-over-end rotation. Under these conditions, Keap1 expressed in cells was shown to be HNEylated by mass-spectrometry and the His-tag pulldown procedure deployed herein yields homogenous His₆-Halo-TeV-Keap1 protein ectopically expressed in cells[13,52,59]. The resin was then washed three times with wash buffers [50 mM HEPES pH 7.6, 150 mM NaCl; and 5 mM imidazole (1st wash), 10 mM imidazole (2nd wash), or 15 mM imidazole (3rd wash)] for 10 min each with end-over-end rotation. His₆-Halo-TEV-Keap1 and binding partners were eluted from the TALON resin with 30 μl elution buffer (50 mM HEPES pH 7.6, 150 mM NaCl, 200 mM imidazole) for 10 min with end-over-end rotation. Eluted proteins were resolved on a 4–20% gradient SDS-PAGE gel (Bio-Rad) and each band was cut into 5 fragments for LC-MS/MS analysis (Cornell Proteomics Core Facility).

**Analysis of Wdr1's relative occupancy on Keap1 following HNE exposure to HEK293T cells.** HEK293T cells transfected with the plasmid encoding either DsRed-IRES-His₆-Halo-TeV-Keap1 or empty vector were cultured for 36 h, and media was changed 24 h post the transfection. The cells were then either treated with 25 μM HNE-alkyne or DMSO vehicle for 1.5 h in serum-free media. Under these conditions, Keap1 expressed in cells was shown to be HNEylated by mass-spectrometry and the His-tag pulldown procedure deployed herein yields homogenous His₆-Halo-TeV-Keap1 protein ectopically expressed in cells[2,3,16]. Cells were then rinsed with 1x PBS and harvested. The cell pellets were then lysed in lysis buffer [50 mM HEPES pH 7.6, 150 mM NaCl, 1% NP-40, 5 mM imidazole, 1X Roche cOmplete, mini, EDTA-free protease inhibitor cocktail, 20 μM PR-619, 200 nM Bortezomib] using three rapid cycles of freeze–thaw. The cell lysates were clarified by centrifugation (21,000 × g, 8 min, 4 °C) and total protein was diluted to 1 mg/ml using cell lysis buffer and incubated with 50 μL TALON resin (Takara Bio) for 1 h at 4 °C with end-to-end rotation. The resin was then washed three times with wash buffers [50 mM HEPES pH 7.6, 150 mM NaCl, 1X Roche cOmplete, mini, EDTA-free protease inhibitor cocktail, 20 μM PR-619, 200 nM Bortezomib; and 5 mM imidazole (1st wash), 10 mM imidazole (2nd wash), or 15 mM imidazole (3rd wash)] for 10 min each with end-over-end rotation at 4 °C. His₆-Halo-TeV-Keap1 and binding partners were eluted from TALON resin using 30 μL elution buffer (50 mM HEPES pH 7.6, 150 mM NaCl, 1X Roche cOmplete, mini, EDTA-free protease inhibitor cocktail, 20 μM PR-619, 200 nM Bortezomib, 200 mM imidazole) over 10 min incubation period with end-over-end rotation at 4 °C. Eluted proteins were resolved on a SDS-PAGE gel and transferred to a PVDF membrane for western blot analysis using indicated antibodies.

**Co-immunoprecipitation of endogenous Keap1 and Wdr1.** General co-IP procedure similar to that previously reported was followed[63]. Briefly, HEK293T cells were treated with either DMSO (–) or HNE (+) (12 μM). After 1.5 h, cells were harvested and washed twice with PBS. Cells were resuspended in lysis buffer (50 mM Tris pH 7.6, 150 mM NaCl, 1% Triton-X-100, 1X Roche cOmplete, EDTA-free protease inhibitor cocktail) and freeze-thawed three times. The lysate was spun down (20,000 × g, 10 min at 4 °C). The supernatant was collected and normalized to 12 mg/ml. 1 mL lysate was precleared with 150 μl settled protein A resin (ThermoFisher, #20334) at 4 °C for 1 h. To the 1 mL precleared lysate was added

75 μl mouse anti-Keap1 bound protein A resin and incubated at 4 °C overnight (the mouse anti-Keap1 bound protein A resin was produced by incubating 1 μg mouse anti-Keap1 with 100 μL settled protein A resin at 4 °C overnight a day prior to the experiment). The resin was pelleted (1000 × g, 1 min) then washed three times for 10 min each in 1 ml of 50 mM Tris pH 7.6, 150 mM NaCl, 0.1% Triton X-100 at 4 °C. The washed resin was added 50 μl 2X Laemmli Buffer containing 50 mM TCEP, and heated to 95 °C for 5 min to elute proteins. The resin was spun down (20,000 × g, 10 min at rt), and the supernatant was used in SDS-PAGE and western blot analyses.

**Generation of lentiviral shRNA-based knockdown HEK293T cell lines.** HEK293T packaging cells (5.5 × 10⁵ cells) were seeded in 6-well plates in antibiotic-free media and incubated for 24 h. The cells were transfected with a mixture of 500 ng packaging plasmid (pCMV-R8.74psPAX2), 50 ng of envelope plasmid (pCMV-VSV-G), and 500 ng of pLKO vector containing the hairpin sequence (Sigma; Supplementary Table 7) using TransIT-LT1 (Mirus Bio) following the manufacturer's protocol. shControl plasmids were obtained from Prof. Andrew Grimson (Cornell University). Eighteen hours post transfection, media were replaced with media containing 20% FBS and incubated for a further 24 h. Media containing virus particles were harvested, centrifuged (800 × g, 10 min), filtered through 0.45 μm filter, and used for infection or frozen at –80 °C for later use.

HEK293T cells (5.5 × 10⁶ cells) in 6-well plates were treated with 1 mL of virus-containing media in a total volume of 6 mL of media containing 8 μg/mL polybrene. After 24 h, media were changed and the cells were incubated for 24 h. Following this period, media were changed to media containing 2 μg/mL puromycin (Corning) and allowed to grow to confluence. Following selection, cells were assayed by western blotting.

**BMDM differentiation analysis.** In vitro differentiation of primary mouse BMCs into BMDMs was performed as described above (see the "Mouse maintenance and culture of bone marrow cells" section). Differentiation was validated by flow-cytometry analysis using previously reported methods and reagents[62]. Briefly, at indicated days post differentiation, 10⁶ cells were incubated with 0.25 μg anti-mouse CD16/32 antibody in 100 μL staining buffer (BioLegend, 429-210) on ice for 10 m. Cells were pelleted by centrifugation, and then resuspended in 100 μL staining buffer containing anti-F4/80 conjugated to AlexaFluor 488 and anti-CD11b conjugated to BV711 and incubated on ice for 20 m in the dark. After being washed twice with staining buffer, cells were analyzed using an Attune NxT Flow Cytometer (AlexaFluor 488: excitation: 488 nm, emission: 530 nm; BV711, excitation: 405 nm, emission: 710 nm). Cells were first gated by FSC-A and SSC-A to determine cells of interest based on size and granularity (typically 55–60%). Further gating was performed by FSC-W and FSC-H, to determine single-cell populations (typically 90–95%). The gates for F4-80 (AlexaFluor 488) and CD11b (BV711) were determined against non-stained cells. Data analysis was performed using FlowJo v10.

**Generation of lentiviral shRNA-based knockdown BMDMs.** HEK293T packaging cells (1.25 × 10⁶ cells) were seeded in T75 flasks in antibiotic-free media and incubated for 24 h. The cells were transfected with a mixture of 2500 ng packaging plasmid (pCMV-R8.74psPAX2), 250 ng of envelope plasmid (pCMV-VSV-G), and 2500 ng of pLKO vector containing the hairpin sequence (Sigma; Supplementary Table 7) using TransIT-LT1 (Mirus Bio) following the manufacturer's protocol. shControl plasmids were obtained from (Sigma or Addgene; Supplementary Table 7). Eighteen hours post transfection, media were replaced with media containing 30% FBS and incubated for a further 24 h. Media containing virus particles were harvested, filtered through 0.45 μm filter, and used for infection.

BMDMs (2.2 × 10⁷ cells) in T75 plates were treated with 4 mL of virus-containing media in a total volume of 24 mL of media containing 4 μg/mL polybrene. After 24 h, media were changed and the cells were incubated for 24 h. Following this period, media were changed to media containing 2.5 μg/mL puromycin (Corning) and allowed to grow to confluence. Following selection, cells were assayed by western blotting.

**Western blotting.** Cells were resuspended in lysis buffer (50 mM HEPES pH 7.6, 1% Triton X-100, 0.3 mM TCEP, 1X Roche cOmplete tablet) and lysed by three cycles of rapid freeze–thaw. Lysates were cleared by centrifugation (20,000 × g, 10 min, 4 °C) and total protein concentration was determined by Bradford assay (against BSA standard). Thirty or fifty micrograms of total protein was loaded per lane, separated by SDS-PAGE, transferred to PVDF, then the membrane was blocked, and incubated with the appropriate antibodies (Supplementary Table 6). Detection was carried out on a ChemiDoc-MP imaging system (Bio-Rad) or a Fusion FX imager (Vilber) using ECL Western Blotting Substrate (Pierce) or SuperSignal West Femto Maximum Sensitivity Substrate (Thermo Scientific). Western blot data were quantitated using the Gel Analysis tool in ImageJ (NIH). Bands of interest were integrated and normalized to the loading control.

**Luciferase reporter assays for measurement of antioxidant response (AR).** These assays were carried out as previously described[13]. Briefly, cells were co-transfected

with a 1:0.025:1:1 mixture of pGL4.37 E364A [(ARE:firefly luciferase) Promega]: pGL4.75, E693A [(CMV:*Renilla* luciferase) Promega]:pCS2 + 8 Halo-TEV-Keap1:pcDNA3 myc-Nrf2 with Mirus 2020 following the manufacturer's protocol for 24 h. Cells were then subjected to T-REX as described above and incubated for a further 18 h. Media were removed and cells were lysed for 15 min in passive lysis buffer (Promega) with gentle shaking and homogenized lysate was transferred to the wells of an opaque white 96-well plate (Corning). Firefly and *Renilla* luciferase activity were measured sequentially on a BioTek Cytation3 microplate reader.

**Luciferase reporter assays**. Cells were co-transfected with a 1:0.025 mixture of firefly luciferase reporter plasmid (Supplementary Table 8): pGL4.75, E693A [(CMV:*Renilla* luciferase) Promega] using Mirus 2020 following the manufacturer's protocol. After 24 h, cells were treated with indicated electrophiles [12 and 24 μM (HNE or HDE); 24 and 48 μM (DMF) for low and high doses, respectively], and incubated for a further 18 h. Cells were lysed and luciferase activity measured as described above.

**Caspase activity assays**. In black 96-well assay plates, lysates from luciferase reporter assays (40 μl per sample) were added to a mixture (150 μl) containing 50 mM HEPES pH 7.6, 0.1% CHAPS, 5 mM DTT, 10 μM Ac-DEVD-AMC. Time-dependent AMC release was measured in fluorescence mode (360 nm excitation; 460 nm emission) on a BioTek Cytation3 microplate reader for 120 min. The activity under these conditions was linear for 90 min. Caspase activity was normalized to the corresponding *Renilla* luciferase signal.

**Growth inhibition assays**. Cells (3000 cells per well for HEK293T cells; 1500 cells per well for MEFs) were seeded in 96-well plates. After 24 h, cells were treated with the indicated molecules at the indicated concentrations and incubated for 48 h. AlamarBlue was added to each well and the cells were incubated for a further 4 h, after which fluorescence (excitation 560 nm; emission 590 nm) was measured using a BioTek Cytation3 microplate reader.

**Cell-cycle analysis**. BMDMs ($8.4 \times 10^5$ cells per well) were seeded in 6-well plates and incubated for 24 h. Cells were harvested, pelleted by centrifugation, and then resuspended in 300 μL Dulbecco's PBS. Chilled EtOH (700 μL) was added dropwise to each sample with regular mixing, and the cells were maintained at 4 °C in dark at least 24 h. Cells were pelleted and washed with 1% BSA in PBS twice, then incubated with 300 μL 50 μg/mL propidium iodide (PI, Sigma-Aldrich, P4864) at room temperature for 30 min with rotation in dark. 30 μL of 10 mg/ml RNase (Qiagen, 1007885) was then added followed by continued rotating for 30 min. Data were acquired using flow cytometry (Attune NxT Flow Cytometer, excitation: 561 nm; emission: 620 nm) and data analysis was performed using FlowJo v10. Cells were first gated by FSC-A and SSC-A to determine cells of interest based on size and granularity (typically 60–75%), from which were further gated by FSC-W and FSC-H to determine single-cell populations (typically 85–95%). The gates among G1, S, and G2/M phases of the cells are indicated in Supplementary Fig. 14b.

**Reporting summary**. Further information on research design is available in the Nature Research Reporting Summary linked to this article.

## Data availability
The RNA-seq data generated in this study have been deposited in the Gene Expression Omnibus (GEO) database under accession code GSE135190. The proteomics data generated in this study have been deposited in the ProteomeXchange database under identifier PXD015481. The remainder of the data generated in this study are provided in Supplementary Tables 1–2, Supplementary Data 1–2, and the Source data file. Source data are provided with this paper.

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

## Acknowledgements

Drs. Chaosheng Luo, Amogh Kulkarni, Yong Chen, Hong-Yu Lin, and Mr. Paul Huang for small-molecule probes; Dr. Florent Duval (Prof. Michele De Palma's laboratory, EPFL) for knowledge transfer regarding BMDM culture; and Ms. Frédérica Schyrr (Prof. Elisa Oricchio's laboratory, EPFL) for murine bone marrow isolation; Prof. Philippe Herbomel (Institut Pasteur) for Tg(lyz:GFP) carmin fish; Dr. Paulina Ciepla for genotyping of and exploratory studies in carmin lines. Staff members at EPFL Center of PhenoGenomics Group (license no. VD3290); Dr. Guillaume Valentin et al. (EPFL zebrafish unit, license no. VD-H23); Nicole McGuire and Prof. Joe Fetcho (Cornell University for zebrafish maintenance: IACUC protocol no. 2017-0055 PI: Aye; no. 2009-0084 PI: Fetcho). Dr. Jen Grenier at Cornell genomics for assistance with RNA-sequence sample processing and data analysis. Dr. Sheng Zhang and staff members at Cornell proteomics for assistance with SILAC-data processing and analysis (NIH SIG grant 1S10OD017992-01, PI: Zhang). American Heart Association predoctoral fellowship (17PRE33670395) (to J.R.P.). Research support: Swiss Federal Institute of Technology Lausanne (EPFL), NCCR Chemical Biology (Swiss National Research Foundation) (to Y.A.). The research was funded by the Swiss National Science Foundation grant number 51NF40_185898 (NCCR Chemical Biology).

## Author contributions

J.R.P. performed mechanistic investigations in zebrafish and cell culture, data analysis, writing (compilation of data, method details, references, and manuscript draft); K.-T.H. performed mechanistic investigations in zebrafish and cell culture, data analysis, writing (compilation of data, method details, references); S.P. discovered the project and performed investigations in zebrafish, data analysis, manuscript proof-editing; Y.Z. performed protein–protein association studies in cell culture, data analysis; S.R. assisted with cell-based reporter assays, data analysis; M.J.C.L. discovered the project and performed investigations in zebrafish, experimental design and data analysis, writing (manuscript draft and critical review); Y.A. conceptualized the project, carried out data analysis, writing (manuscript critical review and editing), supervision, project administration, funding acquisition. All authors assisted with the final proofing of the manuscript.

## Competing interests

A US patent application was filed by Cornell University (former institution of Y.A.) for covalent inhibitors derived from HNE that were discovered using T(Z)-REX.
