## [Peer Review File · Nature Communications]

Reviewers' comments:

Reviewer #1 (Remarks to the Author):

In this manuscript, Poganik et al. identified mediators of a multiple sclerosis drug Tecfidera (DMF) as a Keap1-Wdr1-Cofilin axis, which leads to mitochondrial-targeted neutrophil/macrophage apoptosis. First, they found that Keap1-specific succination induced downregulation of immunity-related genes due to apoptosis of neutrophils and macrophages in zebrafish using T-REX system. Second, by Keap1-pulldown experiments using HEK293T cells and morpholino-based knockdown analysis in zebrafish, they identified that Wdr1 protein increased after Keap1 succination and induced apoptosis in neutrophil/macrophage cells in collaboration with Cofilin. It is a very interesting work, but this reviewer has a number of comments and questions.

1. It is interesting to know whether the protein level of Wdr1 is regulated by ubiquitination or not. Authors should examine the ubiquitination of Wdr1 before and after Keap1 succination.

2. It is not clear that the effect of HNE and DMF are same. Adam et al. (Cancer Cell 20:524-537, 2011) reported that fumarate induced Keap1 succination at Cys23, 151, 273, 288 and 613 of Keap1 and Saito et al (Mol Cell Biol 36:271–284, 2015) showed that Cys151, 273 and 288 were required for the HNE-induced Nrf2 activation. To confirm the results here are indeed Keap1-specific, authors should examine the effects of a C151S&C273W&C288E mutation in Keap1 by Z-REX analysis.

2. Authors showed that the effects of Keap1a and Keap1b were different. Since Cys151, 273 and 288 are somehow different between these two (Li et al. J Biol Chem 283:3248-3255, 2008), authors should discuss about it.

3. It is surprising that non of the reported Keap1-binding proteins, e.g. Nrf2, Nrf1, Cul3, p62, IKKb and so forth, was identified by their Keap1 pull-down analysis. Among these proteins, Cul3 should be dissociated from Keap1 by its succination (Rachakonda et al. Chem Res Toxicol 21:705-710, 2008). Authors should perform control experiments using Cul3 and Nrf2.

4. MO-base knockdown method has been used for all of loss-of-function analyses here. It is good for the screen but not for the confirmation due to its low reliability. For example, authors show that Nrf2b knockdown reduced gstp1-GFP expression, but this was completely different from the results by Timme-Laragy et al. (J Biol Chem 287:4609-4627, 2012), who clearly demonstrated that the expression of endogenous gstp1 was unaffected by Nrf2b knockdown and that the expression of gstp1-GFP was not induced by Nrf2b overexpression. For key conclusions of the paper, authors should use knockout or mutant lines, especially Wdr1.

5. Pirolì et al. (*Mol Cell Proteomics* 18:504–519, 2019) reported that DMF directly succinate Cofilin in neurons and astrocytes. Why did authors ignore this important report? Authors should discuss about this issue.

6. Fuse et al. (*Free Rad Biol Med* 115:405-411, 2018) described that DMF did not induce Nrf2 target genes in zebrafish. Authors should analyze the expression of Nrf2-driven AR genes in DMF-treated zebrafish.

7. There are some papers describing about a zebrafish model for multiple sclerosis, e.g., Kulkarni et al. *Mult Scler Relat Disord* 11:32-39, 2017 and García-Caballero et al. *J Invest Dermatol* 131:1347-1355, 2011. Authors should cite them and discuss more.

8. Above, this reviewer cites 9 papers which seems to be closely related to this manuscript, while authors did cite non of them. Authors should not ignore them.

Reviewer #2 (Remarks to the Author):

Summary: The manuscript by Poganik and coworkers describes the application of the T-REX technology (referred to here as Z-REX when applied to zebrafish) to investigate the mechanism of action of Tecfidera (DMF). While the study purports to focus on DMF, most of the manuscript centers around understanding the mechanism of immunosuppression by a somewhat structurally related electrophile HNE as a surrogate for DMF. The innovation of the T-REX approach is that it limits the electrophile exposure and thus reduces non-specific toxicity. Keap1, one of the putative targets of DMF, is well suited to this approach, as its essentiality has limited the utility of genetic approaches. Using the Z-REX strategy, the authors first conduct RNA-sequencing to identify genes differentially regulated when KEAP1 is presumably modified by HNE, confirming several of these changes by PCR. They then find, using fish expressing RFP/GFP labeled immune cells, that the Z-REX-produced HNE causes a decrease in neutrophil and macrophage numbers in a KEAP1- and HaloTag-dependent manner. The authors then find that immune cells exposed to the Z-REX HNE system show increased apoptosis and that inhibition of caspase-3 restores cell levels. This death was found to not be Nrf2 depending, as indicated by morpholino knockdown studies. The authors then further tease apart the mechanism of action, using the Bax channel blocker (bcb). Bcb modestly protects fish from

the Z-REX induced cell loss. Then the authors switch gears and conduct IP-MS studies to identify KEAP1 binding partners altered by HNE labeling. ~25 hit proteins, identified in two replicate experiments, were then validated in morpholino experiments in fish. While most showed no changes in cell counts, knockdown of the gene *Wdr1* showed a significant protective effect. Heterologous overexpression of *Wdr1* resulted in significantly increased caspase activity. *Wdr1* overexpression also significantly increased Androgen receptor activity. In contrast knockdown did not significantly decrease AR activity. The authors then switch gears again and finally investigate DMF. Fish treated systemically with DMF or HNE, not with the Z-REX system, also show decreases in immune cell counts. Bcb and caspase inhibitors can protect fish from this decrease. Finally, the authors investigate how knockdown of *Keap1*, *Cofilin*, or *Wdr1* impact numbers of immune cells in fish, showing that *wdr1* MOs are protective, whereas *cfl1* and *keap1* are sensitizing. The authors then propose a mechanism whereby HNE or DMF modification of KEAP1 releases *Wdr1*, which leads to association with *cofilin* and intrinsic apoptosis.

Recommendation: While this is an interesting, important, and wide-ranging study, I cannot recommend it for publication in its current form. The manuscript is extremely challenging to decipher and thus the interesting fundamental insights will likely be lost to most readers. Furthermore, the study lacks several controls and mechanistic studies essential for substantiating the proposed mechanism.

Specific comments:

1) In my opinion Figure 4 should be the starting point of the manuscript. In its current form, having the entire introduction about DMF, but most experiments conducted with HNE is confusing and unconvincing. Little data is provided to support that the protein targets labeled by HNE are indeed overlapping with those targets labeled by DMF. No site-of-labeling data is provided nor are *cys-ala* mutants resistant to alkylation. Why is the Z-REX system required when the authors can recapitulate much of the findings with systemic treatments?

2) Does the HNE released by the Z-REX label cysteines in KEAP1? If so, which residues? Could it label cysteines in KEAP1 interactors, such as *Wdr1*? The use of a HaloTag-KEAP1 construct harboring mutation at individual cysteines, as has been done for other investigations of *Keap1* function, would substantially strengthen the manuscript.

3) The key insight of this study is that *Wdr1* interacts with *Keap1* in an HNE/DMF dependent manner and that this interaction is critical for DMF's immunosuppressive activity. While substantial circumstantial evidence is provided to support this claim, key mechanistic experiments are absent. For example, while the authors indicate that *Wdr1* expression level changes modestly upon HNE

treatment (Figure S9A), very little data is provided to functionally validate that these changes are due to dissociation from Keap1. IP-MS experiments are notoriously confounded by false positives and therefore are not sufficient without follow-up studies. IP studies to investigate whether the interactions between Wdr1 and cofilin are HNE/DMF dependent would further support of the proposed mechanism. Immunoprecipitation of Wdr1 by Keap1 and vice versa +/- HNE or DMF or cysteine mutant would confirm that this interaction is specific and dependent on alkylation of one or more cysteine residues.

4) Distinguishing between blockade of proliferation and activation of apoptosis is important for validating the proposed mechanism, particularly given that DMF has been shown to block immune cell activation via alkylation of cysteines in PKCtheta and IRAK4 (Zaro et al 2019). While some data is presented to support the proposed apoptotic mechanism, the addition of immunoblots looking at conventional apoptotic markers (PARP, Caspases, BID, Lamins, etc) and/or quantification of Annexin V/PI, would strengthen the authors claims.

5) Reference to Zaro et al 2019 DOI:10.4049/jimmunol.1801627 should be added.

6) Throughout the manuscript multiple plots are presented which seem to show very similar data. Some appear to be simplified versions of more complex figures (e.g. Fig 3F inset). Some seem to be different data as indicated by statistics that don't match (e.g. Fig2G,H, where the p values change between the left and right panels). Replotting the data in this way is confusing and difficult to decipher at first glance. I would recommend simplifying the data presentation wherever possible to improve manuscript accessibility.

7) Figure 3A would benefit from more descriptive labels to facilitate easy interpretation.

Reviewer #3 (Remarks to the Author):

During the current study, Poganik et al., identify that electrophile engagement of Keap1 by DMF promotes neutrophil and macrophage apoptosis in zebrafish. This apoptosis occurred independent of Nrf2 however, disrupted a Keap1/Wdr1 engagement resulting in enhanced mitochondrial apoptosis. This report adds to the list of mechanisms of action of the blockbuster drug Tecfidera and

adds new knowledge to the field. Although this is a well-done study, there are some necessary experiments to solidify the connection between DMF and human modulation of neutrophil and macrophage apoptosis.

Most of the studies were performed solely in zebrafish. While this is a good model to identify new biochemical interactions and gene functions, none of the findings were confirmed in either primary mammalian neutrophils (human or mice) or even a neutrophil-like cell line. These experiments should be performed to bridge the gap between an interesting observation in zebrafish and concrete effects on these specific cell populations from humans (or mice).

It is understood that primary cells can be difficult to manipulate therefore the authors could employ the human HL60 granulocytic cell line to confirm their results. This cell line can be differentiated into neutrophils, grows well in cell culture and is genetically manipulatable.

The investigators conclude that neutrophils and macrophages in zebrafish are the cellular populations that undergo apoptosis using the Lyz-RFP mice. Since neutrophils, macrophages and monocytic cells can express Lyz, how do the authors know that its neutrophils that are lost/undergo apoptosis? From the manuscript, it was unclear how they distinguished neutrophils from macrophages and monocytes. This should be described at the least.

We are most grateful to the three expert referees for their time and commitment to high standard of reviewing. All of the pertinent questions that were raised are indicative of their knowledge and curiosity. We are only sorry that it took us a considerable length of time to respond fully to their comments and to submit our fully-revised manuscript. Briefly, beyond general restricted access on campus, this revision work was directly affected by, first, prolonged closure of our campus fish unit, and then by the fact that the unit reopened at reduced capacity. These issues caused us significant delays in setting up genotyping and animal crosses necessary to generate requisite embryos for revision experiments. We are grateful to the reviewers and editors for granting us an extended revision period.

Reviewers' comments:

Reviewer #1 (Remarks to the Author):

In this manuscript, Poganik et al. identified mediators of a multiple sclerosis drug Tecfidera (DMF) as a Keap1-Wdr1-Cofilin axis, which leads to mitochondrial-targeted neutrophil/macrophage apoptosis. First, they found that Keap1-specific succination induced downregulation of immunity-related genes due to apoptosis of neutrophils and macrophages in zebrafish using T-REX system. Second, by Keap1-pulldown experiments using HEK293T cells and morpholino-based knockdown analysis in zebrafish, they identified that Wdr1 protein increased after Keap1 succination and induced apoptosis in neutrophil/macrophage cells in collaboration with Cofilin. It is a very interesting work, but this reviewer has a number of comments and questions.

We are thankful for the reviewer's time and thoughtful questions and feedback.

1. It is interesting to know whether the protein level of Wdr1 is regulated by ubiquitination or not. Authors should examine the ubiquitination of Wdr1 before and after Keap1 succination.

As the reviewer is likely aware, ubiquitination can be notoriously difficult to spot, and further a simple observation of ubiquitination does not necessarily mean that ubiquitination is the predominant pathway regulating signaling pathways. That being said, we used several accepted methods to try to observe ubiquitination of Wdr1 and how it changed in a Keap1-dependent manner. We did not observe any clearly discernible/reproducible changes (**Reviewers-only data 1**). We cannot rule out that Wdr1's degradation is either too slow to measure, or that Keap1 is just a stoichiometric inhibitor.

[Redacted]

2. It is not clear that the effect of HNE and DMF are same.

This question partially overlaps with one of the comments from Reviewer 2. We would like to first clarify that our point here is *not* that these compounds are the same, but that they elicit the same phenotypes through the same ‘on-target’ mechanism. We also found that the latter is not the same for another electrophile, HDE (**Fig. 1C, 1F, and 4B**). Such a distinction is important as it requires a lower burden of proof and indeed is a function inherently achievable by two different molecules. Indeed, we often observe in drug–target binding studies/screening that dissimilar molecules can bind ostensibly the same binding sites and elicit the same biological responses. **In our case, we found that DMF and HNE elicit specific loss of neutrophils, require the same 4 proteins to function, and function through the same apoptosis pathway. Our new data further substantiates that these compounds intriguingly share a similar apoptosis mechanism in terms of Bax dependence (new data Fig. S7C-D) and it is different from staurosporine as well.**

Adam et al. (Cancer Cell 20:524-537, 2011) reported that fumarate induced Keap1 succination at Cys23, 151, 273, 288 and 613 of Keap1 and Saito et al (Mol Cell Biol 36:271–284, 2015) showed that Cys151, 273 and 288 were required for the HNE-induced Nrf2 activation.

To confirm the results here are indeed Keap1-specific, authors should examine the effects of a C151S&C273W&C288E mutation in Keap1 by Z-REX analysis.

We generated this construct (**Table S5**) and performed this experiment. We found no decrease in neutrophils (**new data Fig. 1G-H, Fig. S2C**). We believe this is a very important experiment, to rule out off-target labeling, as also proposed by Reviewer 2 (unlikely though such a mechanism be), and to bolster our point. We are grateful for this suggestion.

2. Authors showed that the effects of Keap1a and Keap1b were different. Since Cys151, 273 and 288 are somehow different between these two (Li et al. J Biol Chem 283:3248-3255, 2008), authors should discuss about it.

We are sorry for the misunderstanding. Both Keap1a and Keap1b morphants were amorphic for loss of neutrophils upon DMF bolus dosing (**Figure 5C**). The abovementioned difference between the twain, was observed in the ground state only, and we thus would like to clarify that it has nothing to do with electrophile sensing.

3. It is surprising that non of the reported Keap1-binding proteins, e.g. Nrf2, Nrf1, Cul3, p62, IKKb and so forth, was identified by their Keap1 pull-down analysis. Among these proteins, Cul3 should be dissociated from Keap1 by its succination (Rachakonda et al. Chem Res Toxicol 21:705-710, 2008). Authors should perform control experiments using Cul3 and Nrf2.

We would like to clarify that our experiments were specifically designed to identify *changes in binding*, **not** actual specific binders. This could explain why some of the proteins that the reviewer mentioned were not present in the **table of differential binders (Table S3)**. Furthermore, at endogenous levels, *human* Nrf2 and Nrf1 (the latter **not** believed to be involved in inducible AR function), are tricky to deal with, and are often not observed in MS experiments. For the reviewer’s interest, in the unchanged data set (Table S4, where proteins unchanged upon Keap1 modification are listed in black), the known binder mentioned by the reviewer, p62 (SQSTM1), was detected. The other binders mentioned by the reviewer were not detected in our mass spectrometry data; although we point out that two of the differentially-enriched proteins were also known binders. We would finally point out that one of the possible mechanisms affecting cellular susceptibility to this apoptosis mechanism is Keap1’s interactome, and how that changes across different cells is not well understood. Although out of the scope of this paper, this is something we will be investigating.

As for Cul3, we respectfully state that the study cited above used *N*-iodoacetyl-*N*-biotinylhexylenediamine at 100 μ M. Given the chemodivergence of phenotypes we report here in (HNE vs. HDE), and that *N*-iodoacetyl-*N*-biotinylhexylenediamine is neither of similar chemotype nor concentration to either of the active species we describe, these findings are not relevant to the scope of this manuscript.

4. MO-base knockdown method has been used for all of loss-of-function analyses here. It is good for the screen but not for the confirmation due to its low reliability. For example, authors show that Nrf2b knockdown reduced gstp1-GFP expression, but this was completely different from the results by Timme-Laragy et al. (J Biol Chem 287:4609-4627, 2012), who clearly demonstrated that the expression of endogenous gstp1 was unaffected by Nrf2b knockdown and that the expression of gstp1-GFP was not induced by Nrf2b overexpression.

To further compare against the outcomes we observed in T-REX-assisted on-target HNEylation experiments, we have now extended our validations to studies in HNE-treated primary bone marrow-derived macrophages. Similar to our MO experiments in zebrafish larvae (**Fig. 5A, S15A-C, S16, S17A-C**), we generated and validated multiple shRNA-knockdown lines of primary BMDMs, and derive quite decent levels of consistent outputs from them where these lines show resistant to HNE-induced apoptosis (**new data Fig. 4, Fig. S12A-B**). Our series of orthogonal experiments independently provide evidence that we are on target. Especially for cultured cells, there does not appear to be any strong lines of evidence pertaining to knockout or RNAi being ‘better’, unless these lines be derived from a model organism, in which case off-target effects of knockout can be removed through outcrossing. In that vein, we have additionally used *bax*^{-/-} MEFs, and shown that these are also resistant to HNE- (and also DMF)-induced apoptosis (**new data Fig. S7C-D**). We further showed that staurosporine requires loss of both Bax and Bak to suppress apoptosis, which argues that HNE- and DMF-promoted apoptotic programs are similar.

To unequivocally address the reviewers’ concern regarding knockdowns/use of MOs, and to additionally support the key conclusion, namely, ‘Wdr1 is the necessary mediator of DMF-promoted immune cell apoptosis’, we now further showed that *wdr1*^{-/-} zebrafish embryos are also resistant to neutrophil depletion following DMF treatment (**new data Fig. 5B, S16A,B,C,D**). This resistance is present in both heterozygotes and homozygotes, meaning that *wdr1* is likely a haplosufficient gene.

For the second part of the reviewer’s comment above, related to antioxidant response (AR), importantly, several lines of evidence in our original manuscript, e.g., chemodivergence between HDE and HNE (**Figs. 1A, C, F; 4B**); Nrf2 independence (**Fig. 2E, S5C, S14**), show that AR is a distinct process from what we are discussing in this paper. We do apologize—as the reviewers/editors also related to us—that accessibility/readability of the paper had been an issue, and these aspects likely got buried among the dense data sets. We first clarify that these experiments involving AR are shown mainly to validate MO-knockdown (**specifically Fig. S5A**), because of the paucity of validated human/zebrafish-Nrf2-antibodies. As defined in the corresponding figure legend, the quantitation here refers to a specific region of the zebrafish (the median fin fold of the tail) that is very responsive to electrophiles, and thus reliably reports AR in fish, even when the whole fish (as is typically studied) does not. In fact, the overall levels of GFP (which are highest in the head and are overall poorly-responsive to electrophiles) are not hugely changed upon knockdown (**newly-added data: left inset in Fig. S5A**, referred to as ‘Global GFP signal’), so that observation is consistent with what the Reviewer also proposed. We have also cited the JBC paper that the reviewer referred to (cited ref. # 25).

For key conclusions of the paper, authors should use knockout or mutant lines, especially Wdr1.

We have used the Carmin fish line (cited ref. # 43), which is a Wdr1-knockout line (**new data Fig. 5B, S16A,B,C,D**). These fish show similar behaviors to what we report in morphants, even in the heterozygote, as alluded to above. We have also used MEF-knockout lines (**new data Fig. S7C-D**) to validate the role of Bax in this mitochondria-targeted apoptotic process. We have also substantiated these validations in murine primary macrophages, bone marrow-derived macrophages (BMDMs). As transducing BMDMs for targeted knockdown is an accepted and indeed commercialized strategy, and remains in a biologically-relevant approach, we performed these experiments using multiple shRNA knockdown (**new data Fig. 4, Fig. S12A-B**).

5. Piroli et al. (Mol Cell Proteomics 18:504–519, 2019) reported that DMF directly succinate Cofilin in neurons and astrocytes. Why did authors ignore this important report? Authors should discuss about this issue.

None of our data implicated Cofilin as an electrophile sensor: if Cofilin were sensing HNE leading to loss of neutrophils, we should have seen loss of neutrophils, for instance, in the experiment using split construct expressing Halo and Keap1 as two separate proteins (**Fig. S1A, bottom row** “non-targetable P2A Z-REX construct”), but in fact there was no response (**Fig. 1E-F**). This result is now echoed in the Keap1-mutant data thoughtfully suggested by

this reviewer (**new data Fig. 1G-H, Fig. S2C**). We have used this as an example of where our on-target approach helps rule out collateral damage from genuine sufficient targets.

6. Fuse et al. (Free Rad Biol Med 115:405-411, 2018) described that DMF did not induce Nrf2 target genes in zebrafish. Authors should analyze the expression of Nrf2-driven AR genes in DMF-treated zebrafish.

We would point out respectfully that **HDE also upregulates AR (Fig. 1C, see upregulated AR genes in green) but does not affect neutrophils either under Z-REX (Fig. 1C, 1F) or following bolus dosing (Fig. 4B, right)**. As AR is not directly linked to outputs, performing the suggested experiments would run counter to the logic of the paper. DMF does induce AR effectively in fish; we think that the reviewer would agree that this outcome should have been expected, regardless of the report of Fuse et al. Similarly to the case the reviewer raised above regarding the report by Timme-Laragy et al. (J Biol Chem 287:4609-4627, 2012), we can reconcile our data with that of Fuse et al., since responsiveness of different regions of the fish to canonical AR inducers is not identical. (For instance, please see our expanded **revised Fig. S5A: two plots in inset**). Additionally, we provide below (**reviewers-only data 2**), the data from one of our ongoing projects studying AR-pathway, showing that DMF-treated zebrafish indeed upregulates tail-specific GFP-signal.

[Redacted]

7. There are some papers describing about a zebrafish model for multiple sclerosis, e.g., Kulkarni et al. Mult Scler Relat Disord 11:32-39, 2017 and García-Caballero et al. J Invest Dermatol 131:1347-1355, 2011. Authors should cite them and discuss more.

Thank you for pointing these out to us. We have now added them.

8. Above, this reviewer cites 9 papers which seems to be closely related to this manuscript, while authors did cite non of them. Authors should not ignore them.

We are thankful to the reviewer for these suggestions. A large amount of available literature data were borne from proteomics or studies with limited functional validation and no (accurate) occupancy quantitation, and we thus did not include them in our selected references in the original manuscript. The scope of relevant citations becomes broader still if we were to want to include data derived from treatment with *any* electrophile under *any* conditions/concentrations and cell type. Nonetheless, we hope the reviewer recognizes the fact that all of the relevant citations and additional discussion points have now been integrated in our revised manuscript.

Reviewer #2 (Remarks to the Author):

Summary: The manuscript by Poganik and coworkers describes the application of the T-REX technology (referred to here as Z-REX when applied to zebrafish) to investigate the mechanism of action of Tecfidera (DMF). While the study purports to focus on DMF, most of the manuscript centers around understanding the mechanism of immunosuppression by a somewhat structurally related electrophile HNE as a surrogate for DMF. The innovation of the T-REX approach is that it limits the electrophile exposure and thus reduces non-specific toxicity. Keap1, one of the putative targets of DMF, is well suited to this approach, as its essentiality has limited the utility of genetic approaches. Using the Z-REX strategy, the authors first conduct RNA-sequencing to identify genes differentially regulated when KEAP1 is presumably modified by HNE, confirming several of these changes by PCR. They then find, using fish expressing RFP/GFP labeled immune cells, that the Z-REX-produced HNE causes a decrease in neutrophil and macrophage numbers in a KEAP1- and HaloTag-dependent manner. The authors then find that immune cells exposed to the Z-REX HNE system show increased apoptosis and that inhibition of caspase-3 restores cell levels. This death was found to not be Nrf2 depending, as indicated by morpholino knockdown studies. The authors then further tease apart the mechanism of action, using the Bax channel blocker (bcb). Bcb modestly protects fish from the Z-REX induced cell loss. Then the authors switch gears and conduct IP-MS studies to identify KEAP1 binding partners altered by HNE labeling. ~25 hit proteins, identified in two replicate experiments, were then validated in morpholino experiments in fish. While most showed no changes in cell counts, knockdown of the gene Wdr1 showed a significant protective effect. Heterologous overexpression of Wdr1 resulted in significantly increased caspase activity. Wdr1 overexpression also significantly increased Androgen receptor activity. In contrast knockdown did not significantly decrease AR activity. The authors then switch gears again and finally investigate DMF. Fish treated systemically with DMF or HNE, not with the Z-REX system, also show decreases in immune cell counts. Bcb and caspase inhibitors can protect fish from this decrease. Finally, the authors investigate how knockdown of Keap1, Cofilin, or Wdr1 impact numbers of immune cells in fish, showing that wdr1 MOs are protective, whereas cfl1 and keap1 are sensitizing. The authors then propose a mechanism whereby HNE or DMF modification of KEAP1 releases Wdr1, which leads to association with cofilin and intrinsic apoptosis.

Recommendation: While this is an interesting, important, and wide-ranging study, I cannot recommend it for publication in its current form. The manuscript is extremely challenging to decipher and thus the interesting fundamental insights will likely be lost to most readers. Furthermore, the study lacks several controls and mechanistic studies essential for substantiating the proposed mechanism.

We are grateful for the reviewer's time and input, and positive comments. Respectfully, we do not believe that the paper lacks controls; however, we do cede that there are additional experiments that could be performed to rule out alternative proposals that at least seem sensible to this reviewer. As detailed below, we have carefully addressed these specific concerns. (We would also like to clarify that, in the context of this manuscript, "AR" refers to antioxidant response, not androgen receptor).

Specific comments:

1) In my opinion Figure 4 should be the starting point of the manuscript. In its current form, having the entire introduction about DMF, but most experiments conducted with HNE is confusing and unconvincing.

We have redoubled our effort to significantly revamp accessibility and clarity of the manuscript. We hope that the revised manuscript reflects this major overhaul it had undergone.

We have also extensively considered the reviewer's suggestion to try to commence the manuscript with the data set in Fig. 4A (now Fig 4B in the revised manuscript). However, because we use HNE and HDE (Fig. 1A, 1C, 1F, 4B), and because we also have a unique means to start with a chemically-controlled targeted delivery to a specific protein and track ensuing phenotypic responses, we feel that this starting point focuses the discussion on Keap1, unlike the preponderance of previous studies have been able to do. As the reviewer would equally appreciate, the use of one molecule as a surrogate for another has been a crux of SAR, alkyne tagging, mechanistic studies (including those on DMF) and numerous other studies. Although we are overall against such generalizations, when they can be

directly coupled with precision toolsets, and other negative comparisons [like HDE, which is amorphous for neutrophil loss in Z-REX (**Fig. 1F**)] and under bolus dosing (**Fig. 4B**), together they enable generation of streamlined sets of hypotheses that are experimentally-testable. Such approaches may prove useful in certain contexts, and could bring new and divergent thinking to the field. In the revised manuscript, we have tried our best to improve readability as requested by the reviewer.

We further think that ‘changing gears’ is a necessary component of a multidisciplinary work, which seeks to use several relevant and diverse models to answer a single question. Although perhaps not the most common strategy, we believe such an approach is neither unique to our group, nor unknown to the general reader of this journal.

We finally submit that with new experiments requested by other reviewers, that the manuscript flows better by starting from the general RNA-seq screen by Z-REX than other potential entry points.

Little data is provided to support that the protein targets labeled by HNE are indeed overlapping with those targets labeled by DMF.

We would like to first clarify that we are not interested in generic target-ID, but the biologically-relevant target responsible for changes in neutrophils, which we propose exists at the beginning of the paper. Support that DMF and HNE (*but not HDE*) share a similar pharmaceutically-relevant target in this paper as well as the literature, is indeed legion. Firstly, both DMF¹ and HNE² label Keap1. In our original manuscript, we show that pharmaceutical spectra of DMF and HNE manifest through a similar mechanism, using knockdown/inhibition data in zebrafish (**Fig. 4C, 5A, S15A**) and cells (**Fig. 3C, 3D, 3E, 3G, S10**), all of which agree. We now further show that HNE and DMF elicit apoptosis of MEFs through a similar mechanism that is different to staurosporine (**new data Fig. S7C-D**).

No site-of-labeling data is provided nor are cys-ala mutants resistant to alkylation.

Briefly, out of numerous electrophile-sensor proteins we have studied, human Keap1 with 27 cysteines is uniquely limited for validation via cysteine-mutagenesis, as we have also reported in 4 previous publications³. This is the primary reason why we were not able to use functional Cys-mutants in this work. Reviewer 1 (Query # 2) interestingly suggested quite a unique (Ser,Trp,Glu) triple-mutant-Keap1 previously established as electrophile-sensing-defective but otherwise-active mutant (cited ref. #19 in revised manuscript). We have used this specific Keap1 mutant and T-REX to this mutant does not trigger neutrophil loss (**new data Fig. 1G-H, Fig. S2C**).

Why is the Z-REX system required when the authors can recapitulate much of the findings with systemic treatments?

We think that the reviewer would agree that target-specific data are an integral part of mechanistic investigations. In terms of validating the mechanism, systemic treatments with pleiotropic molecules are not able to show *functional sufficiency* of a target, i.e., precise ramifications of on-target modifications *in vivo*. We have now tried to outline this aspect more clearly in the introduction. We have also critically analyzed the published reports concerning DMF mode-of-action quite recently⁴, where the available data collectively indicated that on-target players with functional necessity and sufficiency were yet to be found.

Critically, identifying the precise consequences of electrophilic modification on a single protein target is a key aspect that cannot be addressed by knockdown studies. Knockdown/knockout can only show pathway sufficiency and are subject to passenger changes. This leaves doubts that Z-REX can fill. More generally, we also point out respectfully that it is much easier to visualize/rationalize the mechanism when most of the cards are on the table, like they are now. At the inception of this work, the ramifications of Keap1-specific labeling and how those events pertain to

¹ Brennan *et al.* *PLoS One* 2015, 10, e0120254

² See, for example: Parvez *et al.* *Nat Protoc.* 2016, 11, 2328–2356; Vila *et al.* *Chem Res Toxicol* 20018, 21, 432–444.

³ Fang, Fu, *et al.* *J Am Chem Soc* 2013, 135, 14496–14499; Parvez, Fu, *et al.* *J Am Chem Soc* 2015, 137, 10–13; Lin *et al.* *J Am Chem Soc* 2015, 137, 6232–6244; Parvez *et al.* *Nat Protoc.* 2016, 11, 2328–2356.

⁴ Invited perspective: Poganik *et al.* Electrophile Signaling and Emerging Immuno- and Neuro-modulatory Electrophilic Pharmaceuticals, 07 February 2020 | <https://doi.org/10.3389/fnagi.2020.00001>

mechanism were unclear. Z-REX allowed us to interrogate these outputs precisely, rather than working from polytropic outputs following bolus dosing of whole animals, which are a major route of much confusion in the field. We believe the derivation of new data based on this work is testament to the utility of the method.

2) Does the HNE released by the Z-REX label cysteines in KEAP1? If so, which residues?

Some of the residues (of human Keap1) labeled under T-REX, in comparison with whole-cell HNE exposure, have been reported,⁵ and that this labeling is fusion-protein-construct-dependent is further illustrated with workflow figure (**Fig. S1A**, bottom row, P2A construct). As T-/Z-REX concept is built on pseudo-intramolecular electrophile delivery,⁶ labeling is transposable from cultured cells to *C. elegans*⁷ and fish.⁸ Residues labeled under such controlled conditions, against those labeled under HNE bulk administration, for human Keap1 have been interrogated with mass spectrometry in our 3 different publications.⁹ Specifically, we went to great lengths to compare side-by-side the identity of residues labeled under T-REX electrophile-limited/controlled conditions (investigating both N-terminal and C-terminal Halo-fused Keap1 constructs, which results in the same residues being labeled) vs. bolus treatment conditions. Thus, respectfully, there is no functional gain in readdressing these in this specific work. Especially given the intrinsic limitations of cysteine mutagenesis for Keap1 protein, as we explained above and also in our published work, residue specificity can only ultimately be used as a guide in this case. Below, as **reviewers-only data**, we show using Click-biotin pulldown method following Z-REX targeted delivery of HNE to Halo-Keap1 in larval fish (in direct comparison with Halo-P2A-Keap1 non-fused construct) that indeed Keap1 is selectively enriched in the fused construct. Comparison with the non-fused P2A construct allowed us to rule out any hidden variables/ potential off-targets in Z-REX.

[Redacted]

Could it label cysteines in KEAP1 interactors, such as Wdr1? The use of a HaloTag-KEAP1 construct harboring mutation at individual cysteines, as has been done for other investigations of Keap1 function, would substantially strengthen the manuscript.

An electrophile unresponsive but otherwise functionally-active mutant of Keap1 is now used and Z-REX on these fish caused no significant change in neutrophils (**new data Fig. 1G-H, Fig. S2C**). These data rule out that second-

⁵ Fang, Fu, *et al. J Am Chem Soc* 2013, 135, 14496–14499; Parvez, Fu, *et al. J Am Chem Soc* 2015, 137, 10–13; Lin *et al. J Am Chem Soc* 2015, 137, 6232–6244; Parvez *et al. Nat Protoc.* 2016, 11, 2328–2356.

⁶ Long *et al. J Am Chem Soc* 2016, 138, 3610–3622.

⁷ Long *et al. Biochemistry* 2018, 57, 216–220.

⁸ Long, Parvez, *et al. Nat Chem Biol* 2017, 13, 333–338.

⁹ Parvez, Fu, *et al. J Am Chem Soc* 2015, 137, 10–13; Lin *et al. J Am Chem Soc* 2015, 137, 6232–6244; Parvez *et al. Nat Protoc.* 2016, 11, 2328–2356.

hand labeling under T-REX to Halo-Keap1 is playing a more dominant role. On a broader level, this observation further substantiates the fact that Keap1 is arguably the best sensors of electrophiles known in vertebrates.

3) The key insight of this study is that Wdr1 interacts with Keap1 in an HNE/DMF dependent manner and that this interaction is critical for DMF's immunosuppressive activity. While substantial circumstantial evidence is provided to support this claim, key mechanistic experiments are absent. For example, while the authors indicate that Wdr1 expression level changes modestly upon HNE treatment (Figure S9A), very little data is provided to functionally validate that these changes are due to dissociation from Keap1. IP-MS experiments are notoriously confounded by false positives and therefore are not sufficient without follow-up studies. IP studies to investigate whether the interactions between Wdr1 and cofilin are HNE/DMF dependent would further support of the proposed mechanism. Immunoprecipitation of Wdr1 by Keap1 and vice versa +/- HNE or DMF or cysteine mutant would confirm that this interaction is specific and dependent on alkylation of one or more cysteine residues.

Data in our original manuscript showed that expressing Wdr1 elevates apoptotic signaling (for instance, **Fig. 3B-C, E**). In terms of our SILAC-based quantitative interactomics analysis of HNEylated Keap1, please also see our responses to Reviewer 1 (Query #3). Based on this reviewer's suggestion, we have now further shown that release of Wdr1 from Keap1 occurs post whole-cell HNE treatment (**new data Fig. S9C**) (under the conditions that HNEylate Keap1 expressed in cells¹⁰).

4) Distinguishing between blockade of proliferation and activation of apoptosis is important for validating the proposed mechanism, particularly given that DMF has been shown to block immune cell activation via alkylation of cysteines in PKCtheta and IRAK4 (Zaro et al 2019). While some data is presented to support the proposed apoptotic mechanism, the addition of immunoblots looking at conventional apoptotic markers (PARP, Caspases, BID, Lamins, etc) and/or quantification of Annexin V/PI, would strengthen the authors claims.

We have now shown PARP cleavage is induced by HNE and DMF and that this is suppressed in Bax knockout MEFs (**Fig. S7D**), as is proliferation suppression (**Fig. S7C**). We make similar experiments in murine primary macrophages, where such PARP cleavage is suppressed by Wdr1 knockdown, following generation of multiple independent knockdown lines in these primary cells (**new data Fig. 4A, Fig. S12A-B**). Our data in the original manuscript showed the actual *loss* of neutrophils in healthy, developing embryos (for instance, **Fig. 2A-B**, showed time-dependent changes in neutrophil counts, following Keap1-specific HNEylation in fish larvae, against controls). Importantly, such an output is not able to arise purely from proliferation inhibition. We link this output to upregulation of cleaved caspase-driven neutrophil-apoptosis in fish (e.g., **Fig 2C, 2F, 2G, 2H, S4C, S6A-B, S7A-D**), an event which is *upstream* of PARP. We also use apoptosis inhibitors to suppress loss of neutrophils (**Fig. 2F, S6A**). (We have also validated the functionality of these inhibitors in their respective assays, **Fig. S10D**).

5) Reference to Zaro et al 2019 DOI:10.4049/jimmunol.1801627 should be added.

Thank you for pointing this out. Our previous manuscript cited the earlier manuscript from the same authors. Based on the reviewer's request, it has now been updated.

6) Throughout the manuscript multiple plots are presented which seem to show very similar data. Some appear to be simplified versions of more complex figures (e.g. Fig 3F inset). Some seem to be different data as indicated by statistics that don't match (e.g. Fig2G,H, where the p values change between the left and right panels). Replotting the data in this way is confusing and difficult to decipher at first glance. I would recommend simplifying the data presentation wherever possible to improve manuscript accessibility.

We indeed added plots of raw data and fold changes (in insets) for the precise reason that it is overall more open for the reviewers/readers to interpret. We strove to provide maximum clarity in defining data in insets by including **Fig. 2D**, which explicitly shows how the fold changes (data in insets) are calculated. These data are processed to be able to statistically evaluate fold changes upon Keap1-HNEylation, the key changes upon which we necessarily build our

¹⁰ Parvez, Fu, *et al. J Am Chem Soc* 2015, 137, 10–13; Lin *et al. J Am Chem Soc* 2015, 137, 6232–6244; Parvez *et al. Nat Protoc.* 2016, 11, 2328–2356.

conclusions. **Fig. 2H**, which the reviewer points out, is a key example of why this analysis is necessary and critical: in this experiment, neutrophil count was significantly affected by Bcb treatment alone. The processed data presented in the inset account for this basal change and allow us to directly compare the effect of Z-REX with or without Bcb treatment. Respectfully, these data are not 'replotted', and thus there is no expectation that p values should be consistent between the raw data and the fold changes presented in insets.

The main issue is that p values do not give a means to directly and fairly compare changes when basal means are variable. If we first assume that means of all basal data sets are equal, we can envision a situation where an experimental procedure, e.g., T-REX, in the control group leads to a large decrease in neutrophils; adjusted $p=0.0001$. However, in the same data set, the same procedure in one knockdown line gives a minimal change, but that change is significant still: adjusted p value = 0.02. Thus, in both instances, a 'significant' change has occurred. As the *magnitude* of p values cannot be compared as a metric of the size of the change (the magnitude of change could be larger for the second data set, but spread could be larger), we are at an impasse.

However, this is not the only information that can be derived from these data as we can also investigate if there was a suppression of change in the knockdown line. As the starting means in each set are the same, the fold changes due to the procedure (e.g., T-REX) can be compared, by asking if the level observed in the first set post-T-REX is different from the level observed in the second post-T-REX. A similar analysis can be carried out when the basal means change. However, in this instance, one needs to directly calculate fold change due to T-REX in each different situation and it is not as intuitively obvious from the raw data alone what has occurred (hence data are recalculated), and the adjusted values are plotted; a transformation that does not simply equate to 'replotting'. As a normalization has occurred, we believe it is the default position to display the data as fold change with specific p values. Hopefully, walking through this analysis with the reviewer will also allow them to apprehend that the p values do not 'change' but a different question is being asked, that otherwise would be unresolvable based on the raw data.

Ultimately, to interpret data from multi-variable systems, more than a first glance is required and for the broader readership with different levels of curiosity, we feel it is important that authors show all their analysis and data interpretation. To mitigate potential confusion, we have now thoroughly ensured that **Fig. S1B** where we outlined our statistical analysis workflow is consistently referred to in all relevant figure legends.

7) Figure 3A would benefit from more descriptive labels to facilitate easy interpretation. We have now expanded the legend and further amended the figures relating to both axes.

Reviewer #3 (Remarks to the Author):

During the current study, Poganik et al., identify that electrophile engagement of Keap1 by DMF promotes neutrophil and macrophage apoptosis in zebrafish. This apoptosis occurred independent of Nrf2 however, disrupted a Keap1/Wdr1 engagement resulting in enhanced mitochondrial apoptosis. This report adds to the list of mechanisms of action of the blockbuster drug Tecfidera and adds new knowledge to the field. Although this is a well-done study, there are some necessary experiments to solidify the connection between DMF and human modulation of neutrophil and macrophage apoptosis.

We are grateful for the reviewer's time and positive evaluations. We also agree with the relevance of the experiments suggested by reviewer.

Most of the studies were performed solely in zebrafish. While this is a good model to identify new biochemical interactions and gene functions, none of the findings were confirmed in either primary mammalian neutrophils (human or mice) or even a neutrophil-like cell line. These experiments should be performed to bridge the gap between an interesting observation in zebrafish and concrete effects on these specific cell populations from humans (or mice).

It is understood that primary cells can be difficult to manipulate therefore the authors could employ the human HL60 granulocytic cell line to confirm their results. This cell line can be differentiated into neutrophils, grows well in cell culture and is genetically manipulatable.

We have now substantiated these findings in murine primary macrophages. Briefly, mouse bone marrow derived macrophages (BMDMs) with Wdr1 knocked down are resistant to PARP cleavage caused by HNE (**new data Fig. 4, Fig. S12A-B**). We also show similar data in MEFs knockouts (**new data Fig. S7C-D**). Please also see our responses to Reviewers 1 and 2 on related questions.

The investigators conclude that neutrophils and macrophages in zebrafish are the cellular populations that undergo apoptosis using the Lyz-RFP mice. Since neutrophils, macrophages and monocytic cells can express Lyz, how do the authors know that its neutrophils that are lost/undergo apoptosis? From the manuscript, it was unclear how they distinguished neutrophils from macrophages and monocytes. This should be described at the least.

We are aware that there is a bit of confusion about the Lyz-marker, especially at this precocious/early developmental stage.¹¹ However, in our original manuscript, we indeed showed specific loss for both *mpeg1* (established macrophage-specific marker; **Fig. S3A, S4A, S4C**) and Lyz-positive cells (neutrophil marker; **Fig. 1D, 1E, 1F, 1G, 1H**). Both of these markers are believed to independently report on neutrophils and macrophages, particularly as fish age, by leading labs using these zebrafish models.¹² Critically, in our hands, these markers also show up separately in age-matched doubly-transgenic fish (**new data Fig. S3C**). Thus, we conclude that Lyz is *not* expressed in macrophages under our conditions. We believe that our newly-acquired data sets in mice primary BMDMs (**new data Fig. 4A, Fig. S12A-B**), MEFs knockouts (**new data Fig. S7C-D**), as well as Wdr1^{-/-} embryos (**new data Fig. 5B, Fig. S16A,B,C,D**), collectively strengthened our diagnostic understanding.

¹¹ Rosowski, *Dis Model Mech* 2020, 12, pii: dmm041889.

¹² Harvie and Huttenlocher, *J Leukoc Biol* 2015, 98, 523–537; Buchan et al. *PLoS One* 2019, 14, e0215592.

REVIEWER COMMENTS

Reviewer #1 (Remarks to the Author):

The revision of the manuscript by authors has addressed the comments raised by this referee. This interesting work seems now suitable for publication in the Nature Communications.

Reviewer #4 (Remarks to the Author):

General comments: The authors have not sufficiently addressed Reviewer 2's most critical comments. The paper is still quite hard to read and lacks sufficient detail for readers who are not very well versed in the authors' previous publications. The narrative meanders in a way that is challenging. It is not due to 'multidisciplinary work' as the authors state but rather a lack of cohesive narrative with clear experiments supporting the claims. Scientifically, the greatest issue lies in the use of HNE as a surrogate for DMF, which is discussed in comment 1 below.

Specific comment 1. The authors still have not addressed this comment. HNE is not a fair surrogate for DMF. DMF is an extremely small molecule and no controls were performed to prove that these molecules function identically with regards to Keap1 or other targets. DMF and HNE targets have to some extent been characterized in two papers that are not referenced by the authors (Blewett et al Science Signaling and Wang et al Nature Methods 2014). It is still unclear why data generated from Z-REX using HNE and not DMF is a superior method for characterization of DMF mechanisms of action.

Specific comment 2. This comment was reasonably addressed.

Specific comment 3. The authors still have not addressed issues with the AP-MS experiments. Additional replicates as well as Co-IP studies would be helpful in this since S9C is supposed to be a pulldown for Keap1 but it was never confirmed by Western blot that Keap1 came down?

Specific comment 4. This comment was sufficiently addressed.

Specific comment 5. Blewett et al Science Signaling should be referenced in addition to Zaro et al.

Specific comment 6. The authors should clarify that the insets are zoom ins of the data adjacent. It is still not clear just looking at the figures.

Specific comment 7. Addressed.

Reviewer #2, #4:

General comments: The authors have not sufficiently addressed Reviewer 2's most critical comments. The paper is still quite hard to read and lacks sufficient detail for readers who are not very well versed in the authors' previous publications. The narrative meanders in a way that is challenging. It is not due to 'multidisciplinary work' as the authors state but rather a lack of cohesive narrative with clear experiments supporting the claims. Scientifically, the greatest issue lies in the use of HNE as a surrogate for DMF, which is discussed in comment 1 below.

Specific comment 1. The authors still have not addressed this comment. HNE is not a fair surrogate for DMF. DMF is an extremely small molecule and no controls were performed to prove that these molecules function identically with regards to Keap1 or other targets. DMF and HNE targets have to some extent been characterized in two papers that are not referenced by the authors (Blewett et al *Science Signaling* and Wang et al *Nature Methods* 2014). It is still unclear why data generated from Z-REX using HNE and not DMF is a superior method for characterization of DMF mechanisms of action.

We have added segues to help the broader readership understand experimental basis/logic and transitions between different result sections. Please see our edits marked in green in the revised manuscript.

As we stated in our original manuscript and previous revision, we surprisingly uncovered through our Z-REX-coupled RNA-Seq screens, that HNE, *but not another related natural electrophile* HDE, can biologically mimic DMF *at that point* only in terms of *on-target* interaction with Keap1. In other words, (1) the preponderance of data in the manuscript indicates that HNE is a fair surrogate for DMF, at least with respect to its interaction with Keap1, particularly with respect to the Keap1-Wdr1 pathway; and (2) we leverage REX technologies to identify a new pathway that has been hidden to other existing approaches. With regard to (1), indeed, it is certainly far from unusual for different pharmacophores to interact with similar binding sites. Contrary to the reviewer's points, this is more likely for 'extremely small (reactive) molecules' as these cannot interact with their targets through ligandable interactions anywhere near as strongly as those with high affinity, and thus these small reactive molecules rather harness intrinsic reactivity (both of themselves and protein nucleophiles). If we consider reactivity preferences, intrinsic reactivities, and transition states for nucleophilic addition, HNE and DMF are far from dissimilar. In fact, as we have written several times, DMF and HNE are much more similar than, for instance, even DMF and iodoacetamide (or other widely-deployed surrogates/profiling agents). Furthermore, it is quite well described that DMF and HNE function similarly with respect to Keap1: both upregulate AR. With regard to (2), ability of Z-REX to zero in on *on-target* effects and the use of non-fused-P2A constructs to rule out off-target consequences, uniquely complements other methods where the latter have failed to produce clear data on phenotypes and *functional sufficiency*. We also further point to the reviewer that all data from Z-REX is validated using bolus DMF dosing, which also agrees with the data resulting from bolus HNE dosing, but not with bolus HDE dosing. Our data also show that the choice of surrogate is important: HDE is not a good surrogate for DMF in Keap1-Wdr1 pathway (although it is for Keap1-Nrf2-AR signaling). In light of all of these data taken together, we believe that Query (1) raised by the reviewer is fully addressed.

Nonetheless, to provide more context, we have now examined this phenomenon in several known stress-responsive pathways (**new data Figure S21**) using numerous luciferase reporter assays. We found that this observed output is a more or less general property of DMF and HNE, versus HDE. However, the underlying explanations for this observation could be manifold and not all linked to Keap1, given the complexity of the systems. Either way, these data do collectively provide credence that DMF and HNE function similarly.

Specific comment 2. This comment was reasonably addressed.

Specific comment 3. The authors still have not addressed issues with the AP-MS experiments. Additional replicates as well as Co-IP studies would be helpful in this since S9C is supposed to be a pulldown for Keap1 but it was never confirmed by Western blot that Keap1 came down?

As per this Reviewer's request, we have now undertaken co-IP studies using the two endogenous proteins. These data show that not only does Wdr1 co-IP with Keap1, but moreover, electrophile treatment diminishes Keap1-Wdr1 association (**new data Fig. S10**: independent biological triplicates). We would respectfully point out that we have aptly demonstrated in 3 publications¹ that this optimized His-tag pulldown procedure deployed in **Fig. S9C**, using the identical line, enriches Keap1 (**stably** expressed) from cultured mammalian cells. For instance, the same procedure enables modification sites of Keap1 enriched from cells treated with various electrophiles to be mapped by mass spectrometry¹, further confirming the validity of experimental approach. Furthermore, this same His-tag pulldown procedure to enrich stably-expressed Keap1 was used in SILAC-interactomics experiments in this

¹ Parvez, S., Long, M.J., Lin, H.Y., Zhao, Y., Haegele, J.A., Pham, V.N., Lee, D.K. & Aye, Y. T-REX on-demand redox targeting in live cells. *Nat Protoc* **11**, 2328-2356 (2016); Lin, H.Y., Haegele, J.A., Disare, M.T., Lin, Q. & Aye, Y. A generalizable platform for interrogating target- and signal-specific consequences of electrophilic modifications in redox-dependent cell signaling. *J Am Chem Soc* **137**, 6232-6244 (2015); Parvez, S., Fu, Y., Li, J., Long, M.J., Lin, H.Y., Lee, D.K., Hu, G.S. & Aye, Y. Substoichiometric hydroxynonylation of a single protein recapitulates whole-cell-stimulated antioxidant response. *J Am Chem Soc* **137**, 10-13 (2015).

current paper (Fig. S8). Thus, in experiments in Fig. S9C, we only looked into answering the key question most relevant to this current work: i.e., to validate our quantitative proteomics data demonstrating that newly-identified binding partner Wdr1 has reduced association with Keap1 in electrophile-treated cells compared to untreated cells. (We provide all N=9 replicates in Source Data File -1). We note finally, that Wdr1 associating with Keap1 in this instance was also endogenous. We have also updated the Main Text and corresponding Method sections with reference to the previous publications.

Specific comment 4. This comment was sufficiently addressed.

Specific comment 5. Blewett et al Science Signaling should be referenced in addition to Zaro et al.

We have cited this paper.

Specific comment 6. The authors should clarify that the insets are zoom ins of the data adjacent. It is still not clear just looking at the figures.

The paper shows only a few “zoom ins” of adjacent data, which are described as “magnification” (e.g., Figure S4C). We believe this is standard in many areas. If we assume that the reviewer was instead referring to insets in quantifications, these are neither zoom ins nor are they simplified versions of more complex data, as proposed by Reviewer 2 (and our response, reproduced below; relevant point highlighted). Insets are, in fact, designed to depict fold change relative to a specific control sample, a parameter that is necessary to compare when differences due to one specific parameter change are affected by a change in another parameter. We and others have published numerous papers with similar analyses, and indeed such concepts are quite commonly applied across the life sciences, forming the basis of concepts such as epistasis, normalization, etc.

6) Throughout the manuscript multiple plots are presented which seem to show very similar data. Some appear to be simplified versions of more complex figures (e.g. Fig 3F inset). Some seem to be different data as indicated by statistics that don't match (e.g. Fig2G,H, where the p values change between the left and right panels). Replotting the data in this way is confusing and difficult to decipher at first glance. I would recommend simplifying the data presentation wherever possible to improve manuscript accessibility.

We indeed added plots of raw data and fold changes (in insets) for the precise reason that it is overall more open for the reviewers/readers to interpret. We strove to provide maximum clarity in defining data in insets by including Fig. 2D, which explicitly shows how the fold changes (data in insets) are calculated. These data are processed to be able to statistically evaluate fold changes upon Keap1-hydroxynonylation, the key changes upon which we necessarily build our conclusions. Fig. 2H, which the reviewer points out, is a key example of why this analysis is necessary and critical: in this experiment, neutrophil count was significantly affected by Bcb treatment alone. The processed data presented in the inset account for this basal change and allow us to directly compare the effect of Z-REX with or without Bcb treatment. Respectfully, these data are not “replotted,” and thus there is no expectation that p values should be consistent between the raw data and the fold changes presented in insets.

The main issue is that p values do not give a means to directly and fairly compare changes when basal means are variable. If we first assume that means of all basal data sets are equal, we can envision a situation where an experimental procedure, e.g., T-REX, in the control group leads to a large decrease in neutrophils; adjusted p=0.0001. However, in the same data set, the same procedure in one knockdown line gives a minimal change, but that change is significant still: adjusted p value = 0.02. Thus, in both instances, a “significant” change has occurred. As the magnitude of p values cannot be compared as a metric of the size of the change (the magnitude of change could be larger for the second data set, but spread could be larger), we are at an impasse.

However, this is not the only information that can be derived from these data as we can also investigate if there was a suppression of change in the knockdown line. As the starting means in each set are the same, the fold changes due to the procedure (e.g., T-REX) can be compared, by asking if the level observed in the first set post-T-REX is different from the level observed in the second post-T-REX. A similar analysis can be carried out when the basal means change. However, in this instance, one needs to directly calculate fold change due to T-REX in each different situation and it is not as intuitively obvious from the raw data alone what has occurred (hence data are recalculated), and the adjusted values are plotted; a transformation that does not simply equate to ‘replotting’. As a normalization has occurred, we believe it is the default position to display the data as fold change with specific p values. Hopefully, walking through this analysis with the reviewer will also allow them to apprehend that the p values do not ‘change’ but a different question is being asked, that otherwise would be unresolvable based on the raw data.

Respectfully, to interpret data from complex systems, more than a first glance is required and for the broader readership with different levels of curiosity, we feel it is important that authors show all their analysis and data interpretation. To mitigate potential confusion, we have now thoroughly ensured that Fig. S1B where we outlined our statistical analysis workflow is consistently referred to in all relevant figure legends.

Specific comment 7. Addressed.

We believe that we have now addressed all of the remaining concerns. We further believe that no comment is in any way detrimental to any paper provided there is fair play and when points are correctly answered, they cease to hold up the publication of good manuscripts. Once again, we thank all of the 4 reviewers for their time and input in genuinely helping to improve our manuscript.

REVIEWERS' COMMENTS

Reviewer #4 (Remarks to the Author):

The authors have sufficiently addressed a large majority of feedback related to the experimental aspects of the manuscript. I still believe the manuscript would benefit from additional textual changes that make it easier to read/follow for those not immediately familiar with the work, but the authors seem resistant to this change. As a reader I feel that it detracts from the work, but as a reviewer it is not a dealbreaker for publication.